**Secondary ice production in summer clouds over the Antarctic coast:**

**an underappreciated process in atmospheric models**

Georgia Sotiropoulou[1,2], Étienne Vignon[3], Gillian Young[4], Hugh Morrison[5,6], Sebastian J. O'Shea[7], Thomas Lachlan-Cope[8], Alexis Berne[3], Athanasios Nenes[1,9]

[1]Laboratory of Atmospheric Processes and their Impacts (LAPI), Ecole Polytechnique Fédérale de Lausanne (EPFL), Lausanne, Switzerland

[2]Department of Meteorology, Stockholm University & Bolin Center for Climate Research, Sweden

[3]Environmental Remote Sensing Laboratory (LTE), EPFL, Lausanne, Switzerland

[4]School of Earth and Environment, University of Leeds, UK

[5]National Center for Atmospheric Research, Boulder, CO, USA

[6]ARC Centre for Excellence in Climate System Science, University of New South Wales, Sydney, Australia

[7]Centre for Atmospheric Science, University of Manchester, UK

[8]British Antarctic Survey, Cambridge, UK

[9]ICE-HT, Foundation for Research and Technology Hellas (FORTH), Patras, Greece

*Correspondence to*: georgia.sotiropoulou@epfl.ch, athanasios.nenes@epfl.ch

**Abstract**

The correct representation of Antarctic clouds in atmospheric models is crucial for accurate projections of the future Antarctic climate. This is particularly true for summer clouds which play a critical role in the surface melting of the ice-shelves in the vicinity of Weddell Sea. The pristine atmosphere over the Antarctic coast is characterized by low concentrations of Ice Nucleating Particles (INPs), which often result in the formation of supercooled liquid clouds. However, when ice formation occurs, the ice crystal number concentrations (ICNCs) are substantially higher than those predicted by existing primary ice nucleation parameterizations. The rime-splintering mechanism, thought to be the dominant secondary ice production (SIP) mechanism at temperatures between -8 and -3$^{\circ}$C, is also weak in the Weather and Research Forecasting model. Including a parameterization for SIP due to break-up (BR) from collisions between ice particles improves ICNC representation in the modeled mixed-phase clouds, suggesting that BR could account for the enhanced ICNCs often found in Antarctic clouds.

The model results indicate that a minimum concentration of about 0.1 L$^{-1}$ primary ice crystals is sufficient to initiate significant break-up, while there is little sensitivity to increasing INPs. The BR mechanism is currently not represented in most weather prediction and climate models; including this process can have a significant impact on the Antarctic radiation budget.

**1. Introduction**

Predictions of Antarctic climate are hampered by the poor representation of mixed-phase clouds over the Southern Ocean and the Antarctic Seas (Haynes et al., 2011; Flato et al., 2013; Bodas-Salcedo et al., 2014; Hyder et al., 2018). Model simulations reveal significant discrepancies in the Antarctic surface radiation budget, associated with cloud biases that are
driven by errors in the representation of the cloud microphysical structure (Lawson and Gettelman, 2014; King et al., 2015; Listowski and Lachlan-Cope, 2017). A correct representation of the cloud radiative impacts largely depends on the parameterization of cloud microphysical processes (Listowski and Lachlan-Cope, 2017; Hines et al., 2019; Young et al., 2019) and precipitation (Vignon et al., 2019), which determine the concentration and
characteristics of liquid drops and ice crystals.

Ice crystals form at temperatures above -38$^{o}$C through heterogeneous nucleation (Pruppacher and Klett, 1997); this means that the presence of insoluble aerosols that act as ice nucleating particles (INPs) is required. However, Antarctica and Southern Ocean are relatively clean regions and INPs are sparse (McCluskey et al., 2018; Schmale et al., 2019;
Welti et al., 2020). Thus it is especially surprising that enhanced ice crystal number concentrations (ICNCs) have been observed in Antarctic clouds (Lachlan-Cope et al., 2016; O'Shea et al., 2017). Secondary ice processes are believed to magnify ICNCs in polar clouds with important implications for the surface radiative balance (Young et al., 2019), yet the underlying mechanisms remain highly uncertain (Field et al., 2017).

The only well-established SIP mechanism that has been extensively implemented in weather forecast and climate models is rime-splintering (Hallett and Mossop, 1974), also known as the Hallett-Mossop process (H-M), which refers to the production of ice splinters after collisions of supercooled droplets with ice particles (Hallett and Mossop, 1974; Heymsfield and Mossop, 1984). This process is effective only in a limited temperature range,
between -8 and -3$^{\circ}$C, and requires the presence of supercooled liquid droplets both smaller and larger than 13 μm and 24 μm, respectively (Mossop and Hallett, 1974; Choularton et al., 1980). However, recent studies have shown that H-M cannot sufficiently explain the

enhanced ICNCs observed in both Arctic (Sotiropoulou et al., 2020) and Antarctic (Young et al., 2019) clouds. While some Antarctic studies (Vergara-Temprado et al., 2018; Young et al., 2019) suggest that the underestimation of ice multiplication in models might be related to uncertainties in the description of the H-M process, we argue that this is likely driven by the fact that almost no models include other SIP mechanisms.

Another SIP mechanism, identified in recent laboratory studies (Leisner et al., 2014; Lauber et al., 2018), is the generation of ice fragments from shattering of relatively large frozen drops. This process however, while very efficient in convective clouds (Korolev et al., 2019), has been found ineffective in polar regions (Fu et al., 2019; Sotiropoulou et al., 2020). This is in agreement with Lawson et al. (2017) and Sullivan et al. (2018a) who have shown that drop-shattering occurs in clouds with a relatively warm cloud base.

Mechanical break-up (BR) of ice particles that collide with each other is another process that results in ice multiplication (Vardiman, 1978; Takahashi et al., 1995) and it operates over a wide temperature range with maximum efficiency around -15°C. Limited knowledge of the BR mechanism comes from few laboratory experiments (Vardiman, 1978; Takahashi et al., 1995) and small-scale modeling (Fridlind et al., 2007; Yano and Phillips, 2011, 2016; Phillips et al, 2017a,b; Sullivan et al. 2018a; Sotiropoulou et al., 2020). To the authors knowledge only two attempts have been made to incorporate this process in mesoscale models (Sullivan et al., 2018b; Hoarau et al., 2018). Specifically, Hoarau et al. (2018) assumed a constant number of fragments ($F_{BR}$) generated per snow-graupel collision in Meso-NH model, while Sullivan et al. (2018b) implemented a temperature-dependent relationship for $F_{BR}$ in COSMO-ART for several types of collisions (e.g. crystal-graupel, graupel-hail, etc), based on the results of Takahashi et al. (1995). Phillips et al. (2017a) recently developed a physically-based description of $F_{BR}$, which is a function of collisional kinetic energy and accounts for the effect of the colliding particles' size and rimed fraction ($\Psi$). While being more advanced than any other parameterization proposed for BR, this scheme has never been implemented in mesoscale models before; it has only been tested in small-scale models for convective clouds (Phillips et al., 2017b; Qu et al., 2020) and Arctic stratocumulus (Sotiropoulou et al., 2020).

Sotiropoulou et al. (2020) recently showed that the observed ICNCs in Arctic clouds within the H-M temperature zone can be explained only by the combination of BR with the H-M process, which results in a 10 to 20-fold enhancement of the primary ice crystals. Based on their results, we postulate that BR may also play a critical role in Antarctic clouds and can potentially explain the discrepancy between the observed and modeled ICNCs in the region

(Young et al., 2019). To test this hypothesis, we implement parameterizations of the BR process in the Morrison microphysics scheme (Morrison et al., 2005) (hereafter M05) in the Weather and Research Forecasting (WRF) model V4.0.1 and examine their influence on the Antarctic clouds observed during the Microphysics of Antarctic Clouds (MAC) field campaign (O'Shea et al., 2017; Young et al., 2019).

## 2. Observations

### 2.1. MAC Instrumentation

The MAC field campaign was conducted in November–December 2015 over coastal Antarctica and the Weddell Sea, with the aim to offer detailed measurements of the microphysical and aerosol properties of the coastal Antarctic atmosphere. MAC included an extensive suite of airborne and ground-based instruments, a detailed description of which can be found in O'Shea et al. (2017). Here we only offer a brief recap of the instrumentation used in this study.

Cloud particle size distributions were derived using the images from a 2D Stereo (2DS, SPEC Inc., USA; Lawson et al., 2006) probe with a nominal size range from 10 to 1280 μm (10 μm pixel resolution). Shattering effects at the probes' inlet were corrected by applying "antishatter" tips (Korolev et al., 2011) and inter-arrival time (IAT) post analysis (Crosier et al., 2011). The 2DS is a single particle instrument, measuring all particles that pass through its sample volume, which depends on particle size and the data integration period. For example, at 300 um, 1 count measured using over a 1-sec averaging window equals to a concentration of 0.27 $L^{-1}$; the uncertainty due to counting statistics is 100%. Total uncertainty is even higher but cannot be quantified.

Aerosol particle measurements of sizes 0.25 to 32 μm were made using the Grimm optical particle counter (GRIMM model 1.109), while a Cloud Aerosol Spectrometer (CAS, DMT; Baumgardner et al., 2001; Glen and Brooks, 2013) measured particles between 0.6 to 50 μm. Following the methodology of Young et al. (2019) and O'Shea et al. (2017), we only consider Grimm measurements of particles with diameter smaller than 1.6 μm in our analysis to avoid including data subject to inlet losses at larger particle sizes. Finally, the aircraft also included instrumentation to measure temperature, turbulence, humidity, radiation and surface temperature (King et al., 2008).

### 2.2 Case study

For our investigations we focus on the MAC case examined in Young et al. (2019), for which they showed that the H-M process, as currently parameterized in WRF, cannot explain the observed ICNCs. Young et al. (2019) utilized measurements from two MAC flights, M218 and M219, combined in one case study; both flights were conducted on 27 November 2015 over the Weddell Sea (Fig. 1): M218 between 15.3-16.7 UTC and M219 between 20.45-22.5 UTC. On that day, a low pressure system persisted over the eastern Weddell Sea, resulting in a southeasterly flow reaching the aircraft with air mass back trajectories from around the low pressure system, towards the Antarctic Peninsula and southern Patagonia (O'Shea et al., 2017).

The temperature and microphysical conditions encountered during these flights are representative of the MAC campaign (see Table 1 in O'Shea et al., 2017, and Fig. S6 in Young et al., 2019). Cloud measurements were collected mainly within the lowest 1.1 km above sea-level (a.s.l.) during both flights and within a temperature range of $\sim$ -9 to -3$^{\circ}$C. The sampled stratocumulus clouds were dominated by supercooled liquid drops, while ice formation occured in isolated ice patches characterized by substantially enhanced ICNCs; the mean (max) ICNCs in these cloud regions were 1.16 (9.03) L$^{-1}$ and 3.33 (87.31) L$^{-1}$ for M218 and M219, respectively. The mean concentration of aerosols with sizes between 0.5-1.6 μm was 0.56 scm$^{-3}$ and 0.41 scm$^{-3}$ (cm$^{-3}$ at standard temperature and pressure) for the two flights. Such low aerosol conditions and concurrent high ICNC concentrations within this temperature range are frequently found in West Antarctic Peninsula (Lachlan-Cope et al., 2016). Moreover, similar cloud droplet concentrations ($N_{drop}$) were measured during both flights (Young et al., 2019): the mean $N_{drop}$ was 82.7 cm$^{-3}$ for M218 and 100.4 cm$^{-3}$ for M219, which are comparable with previous observations from the Antarctic Peninsula (Lachlan-Cope et al., 2016).

## 3. Modeling Methods

### 3.1. Model set-up

This study is conducted with the WRF model (Skamarock et al., 2008), version 4.0.1, by applying the same model set-up as in Young et al. (2019). Two domains with a respective horizontal resolution of 5 km and 1 km are used, where the inner one is two-way nested to the parent domain (Fig. 1). The polar stereographic projection is applied. The outer domain is centered at 74.2$^{\circ}$N, 30$^{\circ}$E and includes 201 × 201 grid points, while the second domain consists of 326 × 406 grids. Both domains have a high vertical resolution with 70 eta levels,

25 of which correspond to lowest 2 km of the atmosphere. The model top is set to 50 hPa. The simulation period spans from 26 to 28 November 2014, 00:00 UTC, allowing for a 24-hour spin up period before the day of interest (27 November). The model timestep is set to 30 (6) sec for the outer (inner) domain, while output data are produced every 30 minutes.

Input data for the initial, lateral and boundary conditions for the simulations are obtained from the European Centre for Medium-Range Weather Forecasting reanalysis (Dee et al., 2011), as recommended by Bromwich et al. (2013). For both shortwave and longwave radiation components, the RRTMG radiation scheme (Rapid Radiative Transfer Model for GCMs) is applied. The Mellor-Yamada-Nakanishi-Niino (MYNN; Nakanishi and Niino, 2006) 2.5-level closure planetary boundary layer (PBL) and surface options are also implemented, in combination with the Noah Land Surface Model (Noah LSM; Chen and Dudhia, 2001), which includes a simplified thermodynamic sea-ice model. Given the short run length, time-varying sea ice concentrations are not utilized. Young et al. (2019) used the Polar WRF V3.6.1 to represent fractional sea-ice, a capability not available in standard WRF V3.6. However, this option has been made available in the more recent V4.0.1 that we use in this study. Following Young et al. (2019), the sea-ice albedo is set to 0.82, with a default thickness of 3 m, and snow accumulation depth on sea ice is allowed to vary between 0.001 m and 1.0 m.

A so-called 'cumulus parameterization' for shallow-convection subgrid processes is not activated in both domains to ensure all cloud processes are represented by the grid-scale microphysics scheme. Note that 5 km is a general upper limit for a convection-resolving resolution (Klemp, 2006; Prein et al., 2015). Cloud microphysics are parameterized following Morrison et al. (2005), hereafter M05. M05 performs well in reproducing Antarctic clouds, resulting in improved representation of the liquid phase and thus the cloud radiative effects compared to less advanced microphysical schemes (Listowski and Lachlan-Cope, 2017; Hines et al., 2019). This bulk microphysics scheme predicts mixing ratios and number concentrations for cloud ice, rain, snow and graupel species. While the mass mixing ratio of cloud water is a prognostic variable, $N_{drop}$ is constant parameter. The default value of the scheme is 200 cm$^{-3}$; here $N_{drop}$ is set to 92 cm$^{-3}$, which is the mean value of M218 and M219 flight measurements (see Section 2.2).

**3.2 Sensitivity Simulations**

A detailed description of the ice formation processes in M05 and the implemented BR parameterizations is offered in Appendix A and B, respectively. We assume that collisions

that include at least one large particle (thus ice-snow, ice-graupel and graupel-snow, snow-snow and graupel-graupel) result in ice multiplication; contribution from collisions between small ice particles (cloud ice) are neglected. In addition to the control (CNTRL) simulation, which corresponds to the default set-up of M05 and accounts only for H-M, we perform seven sensitivity simulations with varying description of $F_{BR}$. We also perform an additional simulation as in CNTRL except with no H-M, and thus no SIP at all, which is referred as NOSIP in the text.

In two sensitivity simulations with active break-up we assume, as in Hoarau et al. (2018), a constant number of fragments generated per collision. This number is constrained by in-situ measurements from the Arctic (Schwarzenboeck et al., 2009) which indicated that one-branch ice-crystals are more common in polar clouds, resulting in ejection of a single fragment after collision with another ice particle. However, this analysis (Schwarzenboeck et al., 2009) included only dendritic crystals with size larger than 300 μm. Based on these results we perform two simulations: FRAG1 assumes all collision types generate one fragment without any size restrictions, while FRAG1siz allows for ice multiplication only if the particle that undergoes fragmentation is larger than 300 μm. Note that because cloud ice with characteristic diameter larger than 250 μm is converted to snow in the M05 scheme, collisions that include cloud ice are assumed to not result in any multiplication in FRAG1siz.

The standard temperature-dependent formula of Takahashi et al. (1995) for $F_{BR}$, applied in Sullivan et al. (2018b), is tested here in the TAKAH simulation. However, Takahashi et al. (1995) used 2-cm hailballs in their experiments, which is an unrealistic set-up. For this reason we perform an additional simulation, TAKAHsc, in which this relationship is further scaled with size (see Appendix B).

Finally, the Phillips parameterization is implemented in three simulations with varying $\Psi$ for the cloud ice/snow particles that undergo fragmentation; $\Psi$ is not predicted in most bulk microphysics scheme, including M05, and thus it is prescribed as a constant. Note that $F_{BR}$ is a function of $\Psi$ only for the ice crystals or snowflakes that undergo break-up, but not for graupel (Appendix B). Graupel is assumed to have $\Psi \geq 0.5$, while the other ice types are characterized by lower rimed fraction. For this reason, we will consider values of $\Psi$ for cloud ice and snow between 0.2 (lightly rimed) and 0.4 (heavily rimed) (Phillips et al., 2017a, b). These simulations are referred as PHIL0.2, PHIL0.3 and PHIL0.4 in the text, where the number indicates the assumed values of $\Psi$ for cloud ice and snow.

## 4. Results

**4.1 BR effect on microphysical properties**

In Fig. 2a the modeled total ice number concentrations (cloud ice + snow + graupel, $N_{isg}$) derived for the region encompassing the 2 MAC flights (Fig. 1) are compared with measurements derived from the 2D Stereo (2DS) probe (see Section 2.1 for details). ICNCs in Fig. 2 are interpolated to match the time resolution of the temperature measurements. Then cloud ice statistics are calculated for $N_{isg} > 0.005$ L$^{-1}$, an indicator for the presence of an ice patch (O'Shea et al., 2017; Young et al., 2019). Moreover, since 2DS cannot resolve the shape (thus cloud phase) of particles smaller than 80 μm, only modeled ice particles with sizes larger than this threshold are considered in Fig. 2, like in Young et al. (2019). While mean and maximum statistics are discussed below, additional statistical metrics (e.g. median and interquartile range) are shown in Fig. S1 (Text S1).

The mean observed $N_{isg}$ for the whole MAC campaign generally fluctuates between 0.5–4.5 L$^{-1}$. The variation in $N_{isg}$ with temperature is somewhat larger for our case study (November 27), as maximum mean concentration goes up to ~6.4 L$^{-1}$ at $T$= -6.5$^{o}$C. Consistently lower concentrations are observed for temperatures $\leq$ -7$^{o}$C, but the temperature statistics are poor for this temperature range as very few observations are available (Fig. 2a). The CNTRL simulation consistently underestimates the mean observations, producing mean $N_{isg}$ ~0.1 L$^{-1}$ over the examined temperature range (Fig. 2a). NOSIP produces similar results to CNTRL, suggesting that the H-M process included in default M05 (CNTRL) is hardly effective at all.

PHIL0.2 and PHIL0.3 also produce similar mean ICNCs to CNTRL (Fig. 2a, b), suggesting that lightly to moderately rimed ice particles do not contribute to ice multiplication through collisional break-up. The higher rimed fraction in PHIL0.4 results in very good agreement with mean observations (Fig. 2a), especially over the whole MAC campaign FRAG1siz also agrees well with mean observations, but when the size restrictions are ignored (FRAG1) the model gives substantial ICNC overestimation. TAKAH simulation also produces unrealistically high mean $N_{isg}$, while TAKAHsc is in closer agreement with observations. The largest deviations from observations for TAKAHsc are observed for temperatures below -7$^{o}$C, where no good measurement statistics are available (see discussion above).

Overall, CNTRL, PHIL0.2 and PHIL0.3 cannot reproduce the observed spectrum (Fig. 2b) and substantially underestimate the frequency of ICNCs larger than 1 L$^{-1}$. PHIL0.4, FRAG1siz and TAKAHsc, however, can successfully reproduce the observed range of values (Fig. 2b), but the relative frequency remains somewhat underestimated. FRAG1 is in closest

agreement with the observed spectrum, while TAKAH often overestimates the relative frequency (Fig. 2b). Maximum ICNCs in FRAG1 and TAKAH are 6403 and 2600 $L^{-1}$, respectively, which are about 70 and 30 times larger that the observed maximum value: 88 $L^{-1}$. This suggests that BR parameterizations that do not account for the impact of size are rather unrealistic. The maximum ICNCs in PHIL0.4, FRAG1siz and TAKAHsc are 174 $L^{-1}$, 150 $L^{-1}$ and 173 $L^{-1}$, which agree to within a factor of two with observations, while they are substantially underestimated in CNTRL (7.8 $L^{-1}$), PHIL0.2 (4.7 $L^{-1}$) and PHIL0.3 (5.2 $L^{-1}$).

Vertical distributions of cloud ice ($N_i$), graupel ($N_g$) , snow ($N_s$) and total ICNC ($N_{isg}$) number concentration are examined in Fig. 3(a-d) for all simulations except those that produce unrealistically large concentrations (FRAG1 and TAKAH). The observed ICNCs are also shown in Fig. 3a and 3c. For consistency with M05, which converts all cloud ice particles with characteristic diameters larger than 250 μm to snow, the same threshold is adopted for splitting the observational dataset in these two ice categories. Graupel concentrations cannot be distinguished in the measurements (hence no 'Nov 27' profile in Fig. 3b); however, the model simulations that are in better agreement with observations (Fig. 2) suggest that these are negligible compared to cloud ice/snow concentrations. Graupel concentrations in Fig. 3b are shown for the whole size spectrum. In contrast, cloud ice (Fig. 3a), snow (Fig. 3c) and total ICNCs (Fig. 3d) include only particles with size larger than 80 μm for consistency with the observations shown in the same panel.

PHIL0.2 and PHIL0.3 produce slightly larger $N_i$ (Fig. 3a) than CNTRL, but reduced $N_g$ (Fig. 3b) values and similar or reduced $N_s$ (Fig. 3c); these mean $N_i$ and $N_s$ profiles are orders of magnitude lower than the observed values. PHIL0.4, FRAG1siz and TAKAHsc produce similar $N_i$ to the observations (Fig. 3a), while $N_s$ is slightly underestimated (Fig. 3c). FRAG1siz is in somewhat better agreement with $N_i$ observations than the other two simulations, especially at heights above 750 m a.s.l (Fig. 3a); this is also reflected in total ICNCs (Fig. 3d). Activating BR generally results in reduction of $N_g$ (Fig. 3b). This decrease is larger than one order of magnitude in the three best performing simulations, compared to CNTRL, however we cannot assess which of these graupel profiles better represents reality. Nevertheless, we can overall conclude that PHIL0.4, FRAG1siz and TAKAHsc result in improved agreement of the vertical distribution of total ICNCs with observations compared to the rest of the simulations (Fig. 3d), including the default set-up of M05. Moreover, cloud ice concentrations (Fig. 3a) are comparable to snow concentrations (Fig. 3c) in these three simulations, in agreement with observations. In contrast, simulations with deactivated or negligible BR result in substantially larger number of snow than cloud ice particles. This

indicates that BR shifts the ice particle spectra to smaller sizes, which results in a more realistic representation of the ice microphysical characteristics.

The simulated liquid water content (LWC) is compared with CAS observations in Fig. 4. All simulations, except TAKAH, produce similar or slightly overestimated mean LWC at temperatures $\leq$ -3.5$^o$C; at -3$^o$C the mean observed values are higher (Fig. 4a). An overestimation of LWC in these runs is more evident in Fig. 4b; the observed spectrum does not include values larger than 0.5 g m$^{-2}$, while the simulated spectra are wider. An exception to this is TAKAH simulation, which underestimates mean LWCs and glaciates clouds at temperatures below -7$^o$C (Fig. 4a), while it produces a narrower LWC spectrum compared to the observed (Fig. 4b). Apart from TAKAH, the remaining simulations produce similar liquid water properties with minor improvements in the runs with reduced LWC values, e.g. in FRAG1 (Fig. 4a). Nevertheless, while the produced range of LWC values in FRAG1 is somewhat closer to the observed, it still underestimates the relative frequency for most of the observed spectrum (Fig. 4b).

**4.2 BR effect on surface radiation**

To examine how deviations in ICNCs affect climate, mean radiative fluxes at the surface and at the top of the atmosphere (TOA) for all model simulations are presented in Table 1. Note that mass mixing ratio fields for all cloud species are provided from the microphysics to the radiation scheme, but no information on droplet and ice effective radius is exchanged.

Increasing BR multiplication has a pronounced impact on shortwave radiation, as it results in decreasing sunlight reflection and thus increasing downward surface radiation (SWD$_{SFC}$). Upward surface radiation (SWU$_{SFC}$) is a function of SWD and thus exhibits similar behaviour. This is due to the fact that increased BR efficiency (Fig. 2) results in decreased liquid water path (LWP) and cloud albedo. The difference between CNTRL and the simulations that improve ICNC representation (PHIL0.4, FRAG1siz and TAKAHsc) fluctuates between 11.9-25.7 W m$^{-2}$ for SWD$_{SFC}$ and 6.7-12.4 W m$^{-2}$ for SWU$_{SFC}$ (Table 1).

Cloud longwave radiative effects are mainly determined by cloud liquid properties, since liquid water is more opaque to longwave radiation than ice particles. However, no substantial differences in mean LWP are indicated for CNTRL (40.1 g m$^{-2}$), PHIL0.2 (33.2 g m$^{-2}$), PHIL0.3 (40.2 g m$^{-2}$), PHIL0.4 (29.1 g m$^{-2}$) and FRAG1siz (30.1 g m$^{-2}$), since LWP values fall within the black body emission range (Stephens, 1978). Optically thinner clouds are produced in TAKAHsc (23.1 g m$^{-2}$), and especially in FRAG1 (8.2 g m$^{-2}$) and TAKAH (3.2 g m$^{-2}$) runs. Note that most simulations, including CNTRL, produce wider LWC spectra

than the observed by overestimating cloud liquid (Fig. 4b). Generally, decreasing liquid content is in better agreement with observations (see section 4.1), suggesting that including the BR process in M05 likely shifts the simulated LWPs towards more realistic values. However, excessive ice multiplication, as in TAKAH, results in unrealistic liquid properties (Fig. 4a) and thus errors in surface radiation.

Pronounced reduction in (LWD$_{SFC}$) is only found for the simulations FRAG1siz, FRAG1 and TAKAHsc, which have a mean LWP well below 30 g m$^{-2}$, the lowest limit of the black body emission range (Stephens, 1978). In all other simulations, the reduction in cloud liquid due to BR is not large enough to alter the cloud emissivity significantly. The upward longwave component (LWU$_{SFC}$) is only slightly affected in all simulations (<~ 1.3 W m$^{-2}$).

Young et al. (2019) showed that underestimation of ICNCs results in significant positive and negative biases in the surface Cloud Radiative Forcing (CRF) over the coastal areas; our results agree with their findings, as CRF biases vary between -78 W m$^{-2}$ and +86 W m$^{-2}$ for the most realistic simulations (Fig. S2, Text S2). Furthermore, the difference between CNTRL and the realistic simulations in upward radiation flux at TOA (Table 1) is also more

pronounced for the shortwave component (SWU$_{TOA}$), fluctuating between 4.7-9.2 W m$^{-2}$, and less significant for LWU$_{TOA}$ (1.4-3.6 W m$^{-2}$). Ultimately, both surface and TOA radiation results indicate that a correct representation of SIP in the atmospheric models is critical for the projection of the energy budget and thus for the future Antarctic climate.

**4.3 Sensitivity to uncertainties in H-M description**

To investigate the interactions between BR and H-M, we compare simulations in which the H-M efficiency is either enhanced or turned off. Young et al. (2019) remove all liquid thresholds from the H-M description, allowing for the process to become active over the whole droplet spectrum. However, this change resulted in very weak ICNC enhancement in

their simulations. Here, we further remove all graupel/snow thresholds from H-M description (Appendix A), which implies that there no size restrictions for the initiation of the process. This modification is applied to CNTRL, PHIL0.3 and PHIL0.4 set-up, resulting in three additional sensitivity tests: CNTRL_NOTHRES, PHIL0.3_NOTHRES and PHIL0.4_NOTHRES, respectively. Furthermore, in addition to NOSIP which corresponds to

CNTRL set-up but without H-M, another two simulations are performed with BR active but H-M completely deactivated: PHIL0.3_NOHM and PHIL0.4_NOHM.

Mean ICNCs in CNTRL_NOTHRES are enhanced by on average a factor of three compared to CNTRL (Fig. 5a). However this simulation underestimates concentrations at

temperatures larger than -7$^{\circ}$C; the mean observed value at this range is 2 L$^{-1}$, while the simulated mean is 0.3 L$^{-1}$. Good agreement between CNTRL_NOTHRES and observations is only achieved at temperatures < -7 °C, where statistical metrics for the two MAC cases are poor (see Section 4.1). While PHIL0.3 did not result in any substantial multiplication, mean ICNCs in PHIL0.3_NOTHRES are 5 times larger. The difference between PHIL0.4 and PHIL0.4_NOTHRES is generally small for temperature warmer than -7$^{\circ}$C, not exceeding a factor of two, but it becomes substantially larger at colder temperatures. However, while the 95$^{th}$ percentiles for CNTRL_NOTHRES and PHIL0.3_NOTHRES are similar and more comparable to observations, PHIL0.4_NOTHRES produces values larger than 10 L$^{-1}$ at all temperatures considered (Fig. 5b).

Excluding the temperature range (<-7 °C) that does not include substantial number of measurements to evaluate model results, mean ICNC observations generally lay between PHIL0.3_NOTHRES and PHIL0.4_NOTHRES in this set of simulations, while CNTRL_NOTHRES produces somewhat lower values (Fig. 5a). However, this set-up overestimates H-M efficiency, as it doesn't include any size limitations, which is not consistent with current knowledge on the H-M mechanism derived from laboratory studies (Hallet and Mossop, 1974; Choularton et al., 1980). Nevertheless, the adapted thresholds are ad-hoc, tuned for different conditions; these should be refined to get a more a realistic H-M effect in polar clouds.

Deactivating H-M completely does not substantially impact the results. This further confirms the fact that the prescribed ad-hoc thresholds prevent the initiation of the process in the studied conditions. Furthermore, it indicates that BR mechanism can explain the observed ICNCs independently of whether H-M is active or not.

## 4.4 Sensitivity to uncertainties in primary ice formation

None of the utilized primary ice nucleation parameterizations are calibrated for the pristine conditions encountered over the high-latitude Southern Ocean, thus it is likely that primary ice formation is overestimated in this case. Moreover, recent studies have suggested the important role of bioaerosols as INPs at the examined relatively warm temperatures (DeMott et al., 2016); this INP type is not accounted for in existing ice nucleating particle parameterizations. To examine how the uncertainty in parameterizations for primary ice affects SIP efficiency, we perform two sets of simulations by dividing or multiplying the efficiency of all primary ice production mechanisms (immersion freezing, contact freezing and deposition/condensation-freezing nucleation) by a factor of 10. Specifically, the first set

with diminished ice nucleation includes CNTRL_INP0.1, PHIL0.3_INP0.1 and PHIL0.4_INP0.1, while the second set with enhanced nucleation consists of CNTRL_INP10, PHIL0.3_ INP10 and PHIL0.4_ INP10.

Decreasing primary ice production by a factor of 10 inhibits BR multiplication (Fig. 6). Note that while maximum $N_{isg}$ in PHIL0.3_INP0.1 and PHIL0.4_INP0.1 is 8.8 L$^{-1}$ and 11 L$^{-1}$ respectively, these simulations produce lower mean values (Fig. 6a) and $N_{isg95}$ (Fig. 6b) than CNTRL_INP0.1 with maximum $N_{isg}$ 5.7 L$^{-1}$. This is partly because a larger number of values in CNTRL_INP0.1 fall below the 0.005 L$^{-1}$ threshold and are not included in the presented mean statistics. Nevertheless, it is clear that the fewer primary ice crystals in all these sensitivity tests result in decreased frequency of ice-ice particle collisions, which is not sufficient to initiate significant BR multiplication.

Increasing primary ice production by an order of magnitude in CNTRL_INP10 still results in underestimated ice concentrations than observed, providing additional evidence for the significant role of SIP in these conditions. The increased concentration of primary ice crystals enhances BR efficiency in PHIL0.3_INP10 compared to PHIL0.3, however the produced mean concentrations still are lower than the observed; $N_{isg95}$ in PHIL0.3_INP10 only slightly exceeds unity (Fig. 6b). In contrast, PHIL0.4_INP10 produces similar mean ICNCs with PHIL0.4. $N_{isg95}$ is also similar at warmer temperatures between two simulations, while larger deviations are observed at temperatures ≤-7$^{o}$C (Fig. 6b). This suggests that increasing concentrations of available ice particles tend to decrease the efficiency of BR mechanism, hence possible overestimations in primary ice production are likely offset by decreasing production of secondary ice.

In summary, the above results indicate that BR cannot be initiated when the available primary ice concentrations are substantially lower than 0.1 L$^{-1}$, which is the mean primary ICNCs produced in NOSIP simulation (Fig. 2a). Yet, INPs over Southern Ocean are often substantially lower (McCluskey et al., 2018; Schmale et al., 2019; Welti et al., 2020). Ice seeding from clouds above the boundary layer was suggested by Young et al. (2019) as a key contributor to the primary ICNC levels for the studied case (see their Supporting Information). Another process that can likely result in optimal conditions for BR to be initiated is aerosol transport from the Antarctic continent, where terrestrial INPs are higher (Vergara-Temprado et al., 2018). Moreover, a combination of these processes and the H-M mechanism, whose efficiency is substantially restricted in the current version of M05, might also provide the necessary concentrations to initiate BR; this was also the case for Arctic stratocumulus conditions in Sotiropoulou et al. (2020). Understanding these interactions

between different processes in the Antarctic region would likely provide insights to the conditions that favor the development of isolated ice patches with substantially high ICNCs within predominantly supercooled liquid clouds. In higher INP conditions, which are likely

less representative of the coastal Antarctic climate, the sensitivity of BR parameterization is expected to be lower.

## 5. Conclusions

Our results indicate that collisional break-up of ice crystals can explain observations of
enhanced ICNCs in coastal Antarctic clouds, but this process requires the presence of ~0.1 L$^{-1}$ primary ice crystals (as produced in NOSIP simulation) for initiation. This likely is a key threshold that can lead the development of isolated ice patches with enhanced ICNCs in predominantly supercooled liquid clouds (Grosvenor et al., 2012; O'Shea et al., 2017). Over the Southern Ocean, when INPs are generally sparse (McCluskey et al., 2018; Schmale et al.,
2019; Welti et al., 2020), such conditions could likely be achieved through ice seeding (as likely happens in the examined case) or through INP transport from the Antarctic continent, where INP concentrations are generally higher (Vergara-Temprado et al., 2018).

Although BR has been observed in polar conditions before (Rangno and Hobbs, 2001; Schwarzenboeck et al., 2009), this mechanism is currently not implemented in most weather
prediction and climate models. The more advanced Phillips et al. (2017a) parameterization produces realistic ICNCs in Antarctic clouds as long as a high rimed fraction is prescribed for the particles that undergo fracture, in agreement with Sotiropoulou et al. (2020). This indicates that our conclusions may not hold for winter clouds in the region, which contain less supercooled liquid water (Listowski et al., 2019) and are less prone to riming. However, for
the studied case, a comparison of vapor deposition rates with riming rates (which include mass changes due to collisions with droplets/raindrops and due to contact/immersion freezing) for CNTRL simulation indicate that these two are on average comparable for cloud ice, while riming rates are substantially larger than vapor deposition rates for snow (not shown). These results suggest that prescribing a high rimed fraction for cloud ice and snow in
M05 is not unreasonable; nevertheless $\Psi$ in reality is highly variable for different temperature and microphysical conditions. More simplified parameterizations also produce improved results as long as the impact of the dependence of $F_{BR}$ on the ice particle size is accounted for.

The very few existing BR descriptions in mesoscale models either do not account for size limitations (Sullivan et al., 2018b) or do not account for all collision types (Hoarau et al.,
2018), which limits their realism. Increasing ICNCs from BR alters significantly the radiative

effects of summer mixed-phase Antarctic clouds; these clouds play a critical role in the surface melting of ice-shelves in the vicinity of Weddell Sea (Gilbert et al., 2020) and thus their accurate microphysical representation in models is of great importance.

**Appendix A: Ice formation processes in M05 scheme**

The standard M05 scheme includes three primary ice production mechanisms through heterogeneous nucleation (immersion freezing, contact freezing and deposition/condensation-freezing nucleation), and one SIP process (H-M).

Immersion freezing of cloud droplets and rain is based on the work of Bigg (1953). This mechanism is active below -4°C and produces a raindrop freezing rate that depends on the degree of supercooling and the number concentration and volume of supercooled drops. The Meyers et al. (1992) description is used for contact freezing, also active below -4°C. The effective diffusivity of the contact nuclei to the drops are estimated from Brownian motion similar to Young (1974): $D_{ap}= R\, T\, (6\, p\, r_i\, N_A\, m\,)^{-1}\, [1 + 0.0737\, T\, (2880P)^{-1}\, r_i^{-1}]$, where $R$ is the universal gas constant, $N_A$ is Avogadro's number, $m$ is the dynamic viscosity of air, $T$ is the temperature, $P$ is the air pressure, and the radius of ice nuclei $r_i$ is assumed to be $1 \times 10^{-7}$ m. The factor in the brackets [] is a correction factor accounting for the mean free path of air molecules relative to the size of the ice nuclei (all units are MKS).

The default parameterization for deposition/condensation-freezing ice nucleation in M05 is from Cooper (1986), which depends only on temperature and is active below -8°C in liquid saturated conditions or when ice supersaturation exceeds 8%. However, the aerosol-aware DeMott et al. (2010) parameterization for heterogeneous nucleation has been shown to compare better with in-cloud ice measurements over the Antarctic Peninsula than Cooper (Listowski and Lachlan-Cope, 2017), although none of these schemes is likely accurate as they have been calibrated for less pristine conditions. Nevertheless, both Cooper and DeMott produce similar primary ice concentrations over the temperature range covered by the observations, but Cooper predicts more primary ice at lower temperatures (<13°C), which might affect the representation of higher-altitude clouds (see Supporting Information in Young et al. 2019). For this reason, we apply the DeMott description in our simulations, where the mean aerosol concentration of particles with sizes between 0.5-1.6 μm for the two flights (0.49 scm$^{-3}$) is used as input (Young et al., 2019). Uncertainty to this formulation is investigated through a number of sensitivity tests (section 4.4).

The H-M parameterization, adapted from Reisner et al. (1998), is based on the laboratory experiments conducted by Hallett and Mossop (1974), who found a maximum of

~350 splinters per milligram of rime generated around -5°C:

$$\frac{dNi_{HM}}{dt} = \rho\, 3.5 \cdot 10^8\, f(T)\, \frac{dm_{rime}}{dt} \quad (1)$$

where $dNi_{HM}/dt$ is the number of new fragments produced at a given timestep, $f(T)$ is the temperature-dependent efficiency of the process, $\rho$ is the air density, and $dm_{rime}/dt$ is the mass production rate of rime on snow or graupel due to accretion of cloud and rain drops. $f(T)$ is 0

for $T <$ -8°C and $T >$ -3°C, 1 for $T =$ -5°C, and increases linearly between these two extremes for $T \geq$ -8°C and T $\leq$ -3°C.

        Furthermore, for H-M to become activated in M05, two conditions must be met: (a) snow (or graupel) mass mixing ratios must be greater than 0.1 g kg$^{-1}$ and (b) cloud liquid (or rain) water mass mixing ratios shoud be greater than 0.5 (or 0.1) g kg$^{-1}$. To achieve a good

agreement between modeled and observed ICNCs for the simulated case, Young et al. (2019) had to remove condition (b) and multiply the H-M efficiency by a factor of 10.

**Appendix B: Parameterizing collisional break-up in M05**

There are three types of ice particles considered in the M05 scheme: small (cloud) ice, snow,

and graupel. Ice multiplication is allowed after cloud ice-snow, cloud ice-graupel, graupel-snow, snow-snow and graupel-graupel collisions. The standard M05 scheme includes a description for collisions between cloud ice and snow to represent the accretion process, following the "continuous collection" approach:

$$\frac{dN_{i_{AC}}}{dt} = \frac{\pi}{4}\rho E_{col}\, \Gamma(b_s + 3)\, a_s\, \frac{N_i N_{0s}}{\lambda_s^{(b_s+3)}} \quad (2)$$

$dN_{iAC}/dt$ is the rate of ice crystal number concentration collected by snow. $N_{0S}$ and $\lambda_s$ are the intercept and slope parameters of the snow size distribution, represented by an inverse exponential function, and $\Gamma$ is the Euler gamma function. $a_s$ and $b_s$ are the characteristic parameters for snow in the fallspeed-diameter relationship (Morrison et al., 2005); $a_s$ includes a density correction factor (Heymsfield et al., 2007). Note that the diameter ($d_i$) and terminal

velocity ($u_i$) of cloud ice particles are considered much smaller than those of snow: $d_i << d_s$ and $u_i << u_s$, so that they are neglected in Eq. (2). $E_{col}$ is the collection (sticking) efficiency between ice particles, set to 0.1; hence, it is assumed that only 10% of cloud ice particles that collide with snow are actually collected. We assume the remaining 90% of collisions result in ice particle break-up, hence the following relationship gives the rate of cloud ice-snow collisions

that contribute to ice multiplication:

$$\frac{dN_{i_{is}}}{dt} = \frac{\pi}{4}\rho(1 - E_{col})\, \Gamma(b_s + 3)\, a_s\, \frac{N_i N_{0s}}{\lambda_s^{(b_s+3)}} \quad (3)$$

In the default M05, collisions between cloud ice and graupel particles are neglected as it is assumed that the collection efficiency of such collisions is negligible. To represent cloud ice-graupel collisions for ice multiplication, we use Eq. (3), but the size distribution and fallspeed parameters of snow are replaced by those for graupel. Moreover, since cloud ice is not collected by graupel particles, we assume that 100% of these collisions result in cloud ice break-up:

$$\frac{dN_{i_{ig}}}{dt} = \frac{\pi}{4}\rho\Gamma\left(b_g + 3\right)a_g \frac{N_i N_{0g}}{\lambda_g^{(b_g+3)}} \quad (4)$$

In the default M05 scheme, collisions between snow and graupel are also neglected because it is assumed that the collection efficiency for such collisions is negligible. For this study, graupel-snow collisions are treated using expressions similar to those for raindrop-snow collisions in M05. These are adapted from Ikawa and Saito (1991) and represent collisions between relatively large precipitation-sized particles:

$$\frac{dQ_{i_{sg}}}{dt} = \pi^2 \rho_s \rho \left|\Delta u_{m_{sg}}\right| \frac{N_{0s}N_{0g}}{\lambda_s^3}\left(\frac{5}{\lambda_s^3\lambda_g} + \frac{2}{\lambda_s^2\lambda_g^2} + \frac{0.5}{\lambda_s\lambda_g^3}\right) \quad (5)$$

$$\frac{dN_{i_{sg}}}{dt} = \frac{\pi}{2}\rho\left|\Delta u_{n_{sg}}\right|N_{0s}N_{0g}\left(\frac{1}{\lambda_s^3\lambda_g} + \frac{1}{\lambda_s^2\lambda_g^2} + \frac{1}{\lambda_s\lambda_g^3}\right) \quad (6)$$

where $\quad \left|\Delta u_{m_{sg}}\right| = \left(\left(1.2u_{ms} - 0.95u_{mg}\right)^2 + 0.08u_{mg}u_{ms}\right)^{1/2} \quad (7)$

and $\quad \left|\Delta u_{n_{sg}}\right| = \left(\left(1.7u_{ns} - u_{ng}\right)^2 + 0.3u_{ng}u_{ns}\right)^{1/2} \quad (8)$

$dQ_{isg}/dt$ and $dN_{isg}/dt$ represent the bulk rates that snow mass and number concentration collide with graupel and contribute to ice multiplication through fragmentation. Corrections in the mass (or number) -weighted difference in terminal velocity $\Delta u_{m_{sg}}$ (or $\Delta u_{n_{sg}}$) of the colliding particles (Eq. 7,8) are adapted from Mizuno (1990) and Reisner et al. (1998), to account for underestimates when $u_{ns} \approx u_{ng}$.

M05 also includes a description for collisions between snowflakes to represent snow aggregation, following Passarelli (1978):

$$\frac{dN_{S_{AG}}}{dt} = \frac{-1108a_s E_{col}}{4\times720}\pi^{\frac{1-b_s}{3}}\rho^{\frac{2+b_s}{3}}\rho_s^{\frac{-2-b_s}{3}}Q_s^{\frac{2+b_s}{3}}N_s^{\frac{4-b_s}{3}} \quad (9)$$

Based on this expression we parameterize the number of snow-snow collisions that contribute to ice multiplication as:

$$\frac{dN_{i_{ss}}}{dt} = \frac{1108a_s(1-E_{col})}{4\times720}\pi^{\frac{1-b_s}{3}}\rho^{\frac{2+b_s}{3}}\rho_s^{\frac{-2-b_s}{3}}Q_s^{\frac{2+b_s}{3}}N_s^{\frac{4-b_s}{3}} \quad (10)$$

Because snow aggregation does not result in any mass transfer, the snow mass involved in these collisions is not calculated by the default M05 scheme. We obtain a description of $dQ_{iss}$

*/dt* by applying the size distribution and fallspeed parameters of snow in the analytical solution for self-collection derived by Verlinde et al. (1990):

$$\frac{dQ_{i_{ss}}}{dt} = \frac{914\pi^2}{48\rho\rho_s}(1 - E_{col})a_s d_s^{b_s+5} N_s^2 \quad (11)$$

To test the consistency of Eq. (10) and (11), which were derived using different methods, we repeated the CNTRL and PHIL0.4 simulations but with the Eq. (9) and (10) replaced by the analytical solution for the change in number concentration from self-collection derived by Verlinde and Cotton (1993). The sensitivity of the results to this modification was found to be insignificant.

Graupel-graupel collisions are also parameterized in a similar manner. Since there is no graupel aggregation (collection efficiency of such collisions is assumed to be negligible), 100% of the collisions are assumed to contribute to break-up:

$$\frac{dN_{i_{gg}}}{dt} = \frac{1108a_g}{4\times720}\pi^{\frac{1-b_g}{3}}\rho^{\frac{2+b_g}{3}}\rho_g^{\frac{-2-b_g}{3}}Q_g^{\frac{2+b_g}{3}}N_g^{\frac{4-b_g}{3}} \quad (12)$$

$$\frac{dQ_{i_{gg}}}{dt} = \frac{836\pi^2}{48\rho\rho_g}a_g d_g^{b_g+5} N_g^2 \quad (13)$$

The value 1108 in Eq. (10) is valid for $b_s$=0.4 (Passarelli, 1978); in M05 $b_s$=0.41 and $b_g$=0.37, thus adapting this value for both snow-snow (10) and graupel-graupel (12) collisions is a reasonable approximation.

Following the methodology of Sullivan et al. (2018b) in TAKAH simulation, the number of fragments generated due to ice-ice particle collisions ($F_{BR}$) is:

$$F_{BR} = 280 \, (T - 252)^{1.2} e^{-(T-252)/5} \quad (14)$$

However, Takahashi et al. (1995) used 2-cm hailballs in their experiments, thus to further include the influence of size in this formulation, we implement a size-scaled expression in TAKAHsc simulation, assuming that $F_{BR}$ depends linearly on D, decreasing to 0 at D = 0:

$$F_{BR} = 280 \, (T - 252)^{1.2} e^{-(T-252)/5} \frac{D}{D_o} \quad (15)$$

where *D* (in meters) is the size of the ice particle that undergoes fracturing and $D_o$=0.02 m, the size of haiballs used by Takahashi et al. (1995) .

The Phillips et al. (2017a) parameterization allows for varying treatment of $F_{BR}$ depending on the ice crystal type and habit.

$$F_{BR} = \alpha A \left(1 - exp\left\{-\left[\frac{CK_o}{\alpha A}\right]^\gamma\right\}\right) \quad (16)$$

$$\text{where}: \ K_o = \frac{m_1 m_2}{m_1 + m_2}\left(\Delta u_{n_{12}}\right)^2 ,$$

$$\psi = 3.5\times10^{-3}$$

$$a = \pi D^2$$

$m_1$, $m_2$ are the masses of the colliding particles and $\Delta u_{n12}$ is the difference in their terminal velocities. The correction applied in Eq. (8) is also adapted here to account for underestimates when $u_{n1} \approx u_{n2}$. $D$ (in meters) is the size of the smaller ice particle which undergoes fracturing and α is its surface area. The parameterization was developed based on particles with diameters 500 μm $< D <$ 5 mm, however Phillips et al. (2017a) suggest that it can be

used for particle sizes outside the recommended range as long as the input variables to the scheme are set to the nearest limit of the range. $C$ is the asperity-fragility coefficient and $\psi$ is a correction term for the effects of sublimation based on the field observations by Vardiman (1978). For cloud ice-snow, cloud ice-graupel, snow-graupel and snow-snow collisions:

$$A = 1.58 \cdot 10^7 (1 + 100\Psi^2)\left(1 + \frac{1.33 \cdot 10^{-4}}{D^{1.5}}\right),$$

$$\gamma = 0.5 - 0.25\Psi,$$
$$C = 7.08 \times 10^6 \psi$$

The above parameters adapted from Phillips et al. (2017a) concern planar crystals or snow with rimed fraction $\Psi < 0.5$ that undergo fracturing: $\Psi \leq 0.2$ corresponds to lightly rimed particles, while $\Psi \approx 0.4$ represents highly rimed crystals/snow. The choice of the ice habit is based on particle images collected during the MAC flights, which indicate the presence of

needles and planar particles (O'Shea et al., 2017); needles are often considered secondary ice (Field et al., 2017). However, the M05 scheme does not explicitly consider habit and assumes spherical particles for all processes except sedimentation, for which the fallspeed-diameter relationships are for non-spherical ice.

For graupel-graupel collisions the parameters implemented in Eq. (16) are somewhat

different (Phillips et al., 2017a):

$$A = \frac{a_o}{3} + \max\left(\frac{2a_o}{3} - \frac{a_o}{9}|T - 258|, 0\right)$$
$$\gamma = 0.3,$$
$$C = 6.3 \times 10^6 \psi$$

Finally, an upper limit for the number of fragments produced per collision is imposed, set to $F_{BR_{max}} = 100$; this is the same for all collision types (Phillips et al., 2017a).

We estimate the production rate of fragments for cloud ice-snow collisions and cloud ice -graupel collisions using Eq. (3) or (4) and one of the proposed formulations for $F_{BR}$ above: $\frac{dN_{i_{is}}}{dt} F_{BR}$ and $\frac{dN_{i_{ig}}}{dt} F_{BR}$. For both of these collision types we assume that the cloud ice particles undergo break-up and the new smaller ice fragments remain within the same ice particle category. For snow-graupel collisions, where the snow particle is assumed to undergo

fracturing, the production term $\frac{dN_{i_{sg}}}{dt} F_{BR}$ is added to the cloud ice category. In this case mass transfer from the snow to the cloud ice category also occurs, but according to Phillips et al. (2017a) this is only 0.1% of the snow mass that collides with graupel (5). Snow-snow and graupel-graupel collisions are handled in the same way: $\frac{dN_{i_{ss}}}{dt} F_{BR}$ and $\frac{dN_{i_{gg}}}{dt} F_{BR}$ are added to the cloud ice number equation, while 0.1% of $\frac{dQ_{i_{ss}}}{dt}$ (11) and $\frac{dQ_{i_{gg}}}{dt}$ (13) is added to the

corresponding mass equation.

**Code and data availability:** MAC data are available at https://catalogue.ceda.ac.uk/uuid/da17dab196f74d64af3ccbc35624027b. The modified Morrison scheme is available upon request


**Competing interests:** The authors declare that they have no conflict of interest.

**Author contribution:** GS and AN conceived and led this study. EV helped with the model configuration and set-up, and provided Fig. 1. GY provided the observations and the model
set-up for the MAC case. SJO post-processed MAC data. GS implemented the BR parameterizations, performed the WRF simulations, analyzed the results and, together with AN, led the manuscript writing. All authors contributed to the scientific interpretation, discussion and writing of the manuscript.

**Acknowledgements:** GS and AN acknowledge support from Laboratory of Atmospheric Processes and Their Impacts (LAPI) at the École Polytechnique Fédérale de Lausanne (EPFL) the project IC-IRIM (project ID 2018-01760) funded by the Swedish Research Council for Sustainable Development (FORMAS), the project PyroTRACH (ERC-2016-COG) funded from H2020-EU.1.1. - Excellent Science - European Research Council (project ID 726165)
and the project FORCeS funded from Horizon H2020-EU.3.5.1. (project ID 821205). EV and AB acknowledge the financial support from EPFL-ENAC through the LOSUMEA project. The National Center for Atmospheric Research is sponsored by the U.S. National Science Foundation. GY acknowledges support from the UK Natural Environment Research Council (grant no.: NE/R009686/1). We are also grateful to MAC scientific crew for the observational
datasets used in this study.

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

**Tables:**


**Table 1:** Mean modeled downward and upward shortwave (SWD$_{SFC}$, SWU$_{SFC}$) and longwave (LWD$_{SFC}$, LWU$_{SFC}$) surface radiation, along with upward shortwave and longwave (SWU$_{TOA}$, LWU$_{TOA}$) radiation at the top of the atmosphere, during flights M218 and M219. Model results are averaged over the dashed rectangular area in Fig. 1.


| Simulation | SWD$_{SFC}$ (Wm$^{-2}$) | SWU$_{SFC}$ (Wm$^{-2}$) | LWD$_{SFC}$ (Wm$^{-2}$) | LWU$_{SFC}$ (Wm$^{-2}$) | SWU$_{TOA}$ (Wm$^{-2}$) | LWU$_{TOA}$ (Wm$^{-2}$) |
|---|---|---|---|---|---|---|
| CNTRL | 323.9 | 182.1 | 244.3 | 304.6 | 255.8 | 218.4 |
| PHIL0.2 | 328.6 | 184.5 | 244.1 | 304.6 | 254.8 | 218.5 |
| PHIL0.3 | 322.3 | 181.0 | 247.4 | 305.3 | 256.6 | 217.9 |
| PHIL0.4 | 339.7 | 190.8 | 243.3 | 304.9 | 251.1 | 219.8 |
| FRAG1 | 354.1 | 198.6 | 236.7 | 303.8 | 246.9 | 221.5 |
| FRAG1siz | 335.7 | 188.8 | 244.0 | 304.6 | 250.5 | 220.7 |
| TAKAH | 365.9 | 206.5 | 229.8 | 303.3 | 242.5 | 221.2 |
| TAKAHsc | 349.5 | 194.5 | 237.0 | 304.2 | 246.6 | 222.0 |

**Figures:**

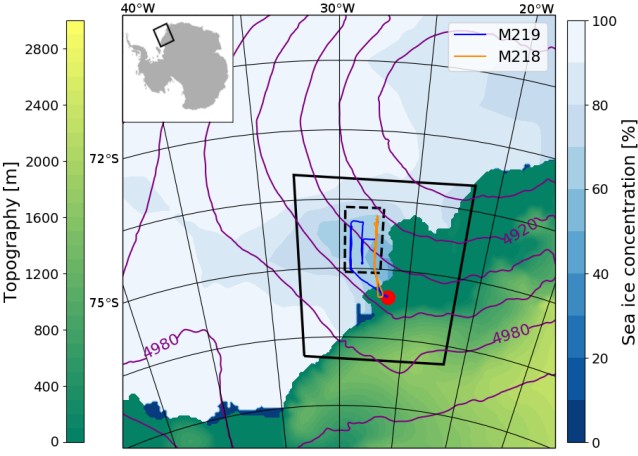

**Figure 1:** Map of Antarctic domains. Colors indicate terrain heights (green to yellow) and sea-ice concentrations (blue to white), whereas the purple contours correspond to 500 hPa geopotential heights from the CNTRL simulation at 18:00 UTC, 27 November 2015. The black solid line delimits the 1-km horizontal grid spacing domain, while the dashed one outlines the subset of the nest used for direct comparison with the aircraft data. Orange and blue lines indicate the flight tracks, while the red circle represents Halley station. The small figure in the top right corner indicates the location of the 1-km horizontal grid spacing domain relative to the Antarctic continent.

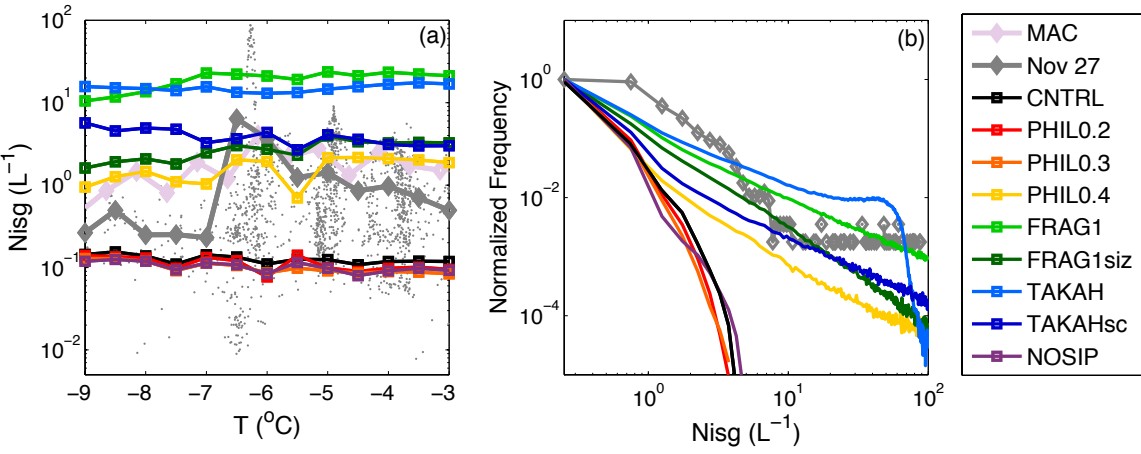

**Figure 2:** (a) Mean ice number concentrations (cloud ice + snow + graupel, $N_{isg}$) as a function of temperature for the whole MAC campaign (pink), our case study (grey) and the eight model simulations. Grey dots indicate point observations. (b) Relative frequency distribution of $N_{isg}$, binned in 0.5 L$^{-1}$ intervals, scaled with maximum frequency. Ice properties are calculated for particles > 80 µm and for $N_{isg}$ > 0.005 L$^{-1}$ within the lowest 1.5 km a.s.l.

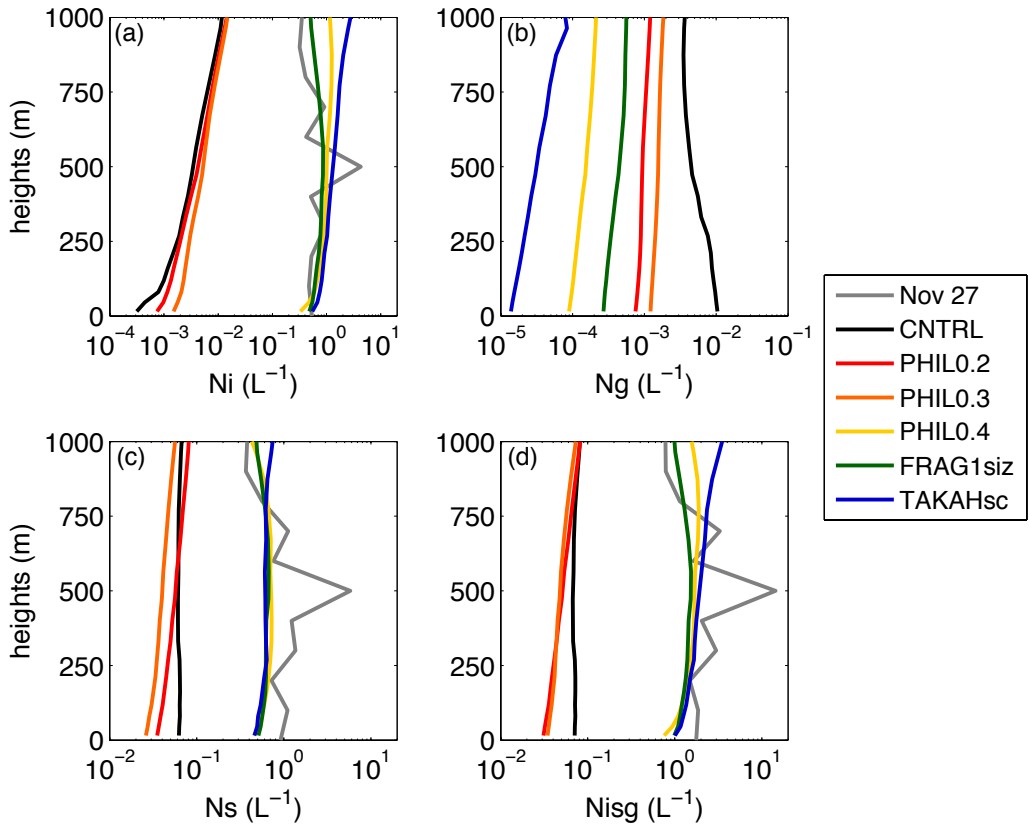

**Figure 3:** Mean vertical profiles of number concentrations of modeled (a) cloud ice, (b) graupel, (c) snow and (d) total ICNCS for six simulations. Grey lines represent measured concentrations with diameters (a) smaller and (c) larger than 250 μm. Graupel concentrations cannot be distinguished in the measurements (hence no grey profile in panel b). Ice properties from the model are calculated for $N_{isg}>0.005$ L$^{-1}$. For consistency with observations, only particles with sizes > 80 μm are included in the modeled profiles in panels (a), (c) and (d).

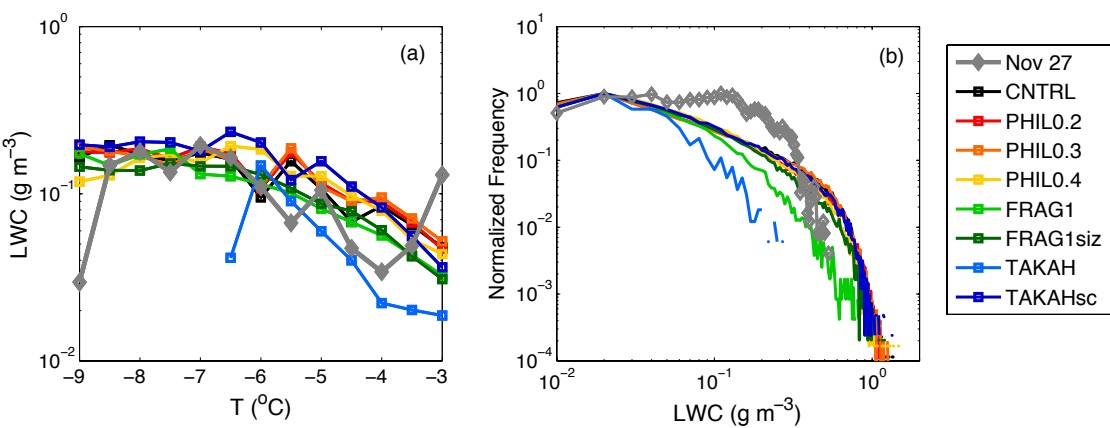

**Figure 4:** (a) Mean liquid water content (LWC) as a function of temperature for our case study (grey) and the eight model simulations. (b) Relative frequency distribution of LWC, binned in 0.01 g m$^{-3}$ intervals, scaled with maximum frequency. Only values greater than 0.01 g m$^{-3}$ within the lowest 1.5 km a.s.l. are included in the analysis.





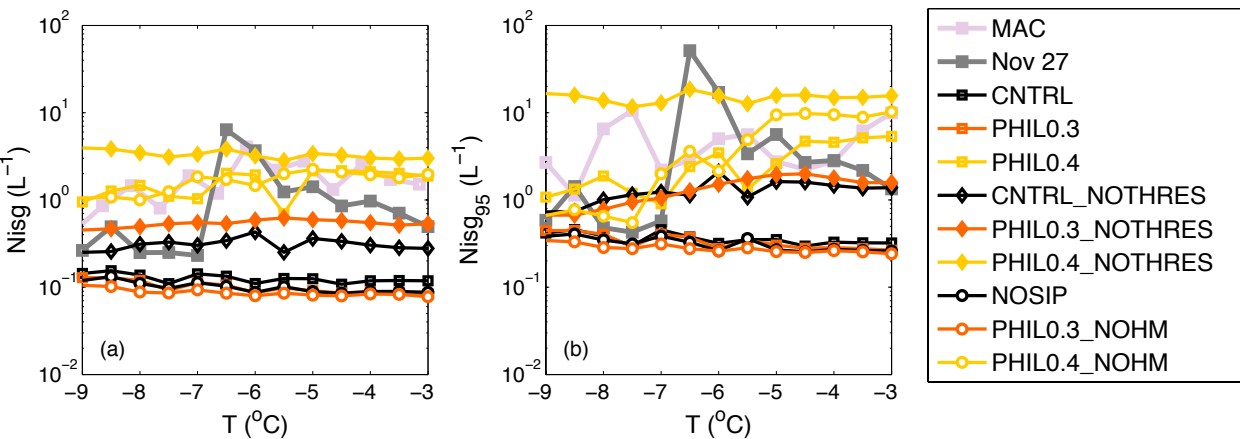

**Figure 5:** Total ice number concentrations ($N_{isg}$) for particles > 80 μm as a function of temperature for the while MAC campaign (pink), our case study (grey) and the nine sensitivity simulations with varying treatment of the H-M process (see section 4.3). Mean values and the 95th percentile are shown in panels (a) and (b), respectively.

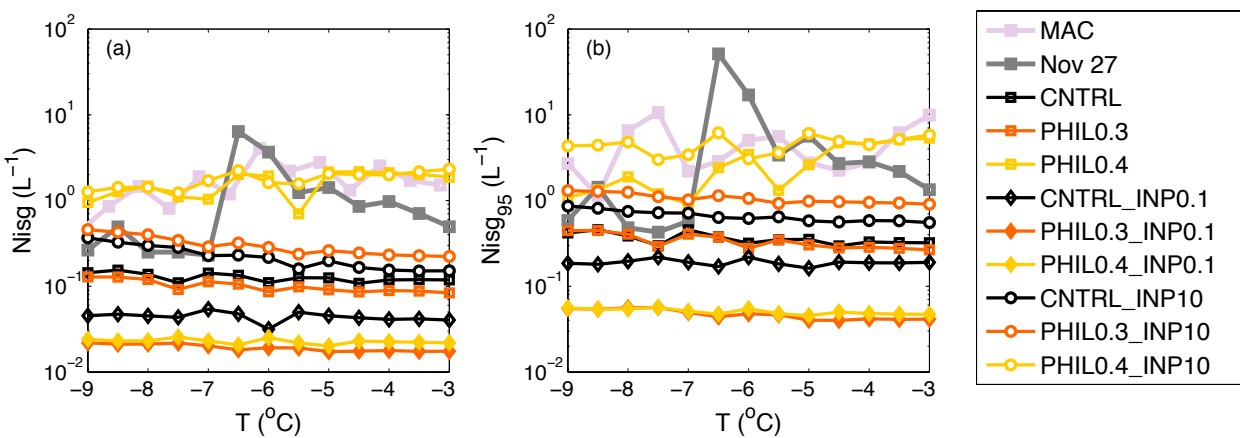

**Figure 6:** Same as in Fig. 5 but for sensitivity simulations with varying INP conditions (see
section 4.4)



