# Peer review of "Secondary ice production in summer clouds over the Antarctic coast: an underappreciated process in atmospheric models"

_Atmospheric Chemistry and Physics, 2020_

## Referee Comment (RC1) · Anonymous Referee #1 · 22 Jun 2020

General comments: The study simulates an Antarctic cloud over the coastal Antarctic and the Weddell Sea in November-December 2015. Secondary ice production from the break-up of collisions between ice particles is added to Weather and Research Forecasting model. The model simulated results are compared to extensive measurements from airborne and ground-based instruments. Their results indicated break-up of collisions between ice particles could account for enhanced ice number in the pristine Antarctic atmosphere, and these results are insensitive to uncertainties in primary ice production. I think the study will be publishable after the following comments are

addressed.

specific comments: 1. My main concern is the experimental design. The control simulation is stated as using the default Morrison scheme (Line 204: "Additionally to the control (CNTRL) simulation, which corresponds to the default set-up of M05"). The Morrison scheme is described in the literature that includes different types of secondary ice production in it, i.e. the rime splintering (H-M) during snow accretion cloud droplets and snow accretion raindrops. Therefore, either I have misinterpreted and some additional description of the control model configuration is needed, or I think we need an additional set of control runs that have no secondary ice production processes included Until there is a clean experiment with no secondary ice production processes it is difficult to interpret the statements about the impact of secondary ice production.

2. My first concern is about "separation size between ice and snow" The authors mentioned in the manuscript Line 215: "Note that since the separation size between ice and snow in the M05 scheme is 125 $\mu$m, collisions that include cloud ice do not result in any multiplication in FRAG1siz." The reader may be confused, does Morrison scheme really has a size separation between cloud ice and snow? Does this mean the model does not have cloud ice larger than 125 $\mu$m, and no snow particles with a diameter smaller than 125 $\mu$m? Morrison scheme is documented in the literature that includes a threshold size (125 $\mu$m) for the cloud ice autoconversion process. But this does not mean a size separation between could ice and snow. Based on this, when comparing simulated snow and graupel with observation, does modeled snow and graupel only considering particles larger than 80 $\mu$m? Or only modeled cloud ice has this threshold in size.

3. Related to the second concern, the third concern is related to the comparison between observation and model results. The authors mentioned that "consistency with M05, the threshold size separating measured cloud ice from snow is set to 125 $\mu$m" Usually, when comparing model ice with observation, we added different types of modeled solid particles together, then using the total mass and total number to compare

with the observed IWC and ice number concentration. Because it is hard to tell cloud ice from snow in observation data. I suggested using a similar method when compare the modeled ice number with observation.

4. The fourth concern is related to parameterization from Phillips et al. (2007) In the sensitivity test, Line 255, "These simulations are referred to as PHIL0.2, PHIL0.3 and PHIL0.4 in the text, where the number indicates the assumed values of $\Psi$". In the bulk microphysical scheme, snow is referred to as dry snow, with smaller density (prescribed and fixed in the scheme), graupel is rimed ice with larger density. Setting the same rimed fraction for different collisions is not consistent with the assumptions in the microphysics scheme. I suggested the uses different rimed fraction for snow-snow and graupel-graupel collision. The rimed fraction could be changed in the sensitivity test, but in one simulation, the rimed fraction for the collision between snow-snow should not larger than rimed fraction for the collision between graupel-graupel.

5. The last one is related to radiation Does cloud microphysical properties couple with radiation transform code? It is interesting to see the model has a larger ice number and IWC, after the implementation of the secondary ice process, but the longwave radiation does not change accordingly. How about the effective radius of the ice particle? Does it change after the model has secondary ice production in it?

technical corrections 1. Line 59 "Lachlan-Cope et al., 2016; Wexx et al.,". Wexx –> Wex? 2. Line 96 be –> been 3. Code and data availability: the link authors provide does not link to the measurement data, please upload the data.

Please also note the supplement to this comment:
https://www.atmos-chem-phys-discuss.net/acp-2020-328/acp-2020-328-RC1-supplement.pdf

---

## Referee Comment (RC2) · Anonymous Referee #3 · 22 Jun 2020

**General Comments**

This is a good paper overall, in its seeking to explore the role of different secondary ice formation mechanisms active under conditions present in clouds over the focused region near Antarctica. To the extent that I can tell, the findings with regard to BR parameterizations are solid and enlightening regarding the necessary conditions (especially heavy riming) for it to occur and the need for formulations of BR that account for the requisite size of colliding particles. This paper was less forward and unsatisfying in conveying that end-to-end prediction of secondary ice formation actually still eludes

the community. This is my inference from this paper and from the literature of the last few years represented by some of the team on this paper. This relates to the assumptions on primary nucleation, and the concentrations of ice that exists outside of regions where strong SIP ensues. There is still a gap here, one that is not really discussed, although this study seems to close it some. By this I mean that the authors have proposed to explain how ice concentrations go from values of 0.1 per liter to ones enhanced up to 100 times, and under conditions that the Hallett-Mossop process cannot explain such enhancement. The results cannot explain the inconsistency between recent INP measurements over the Southern Ocean or near Antarctica (e.g., McCluskey et al., 2018; Schmale et al., 2019; Welti et al., 2020) and the fact that Litkowski et al. (2017) required use of the DeMott et al. (2010) parameterization (with qualifications about its likely inapplicability to the region) in order to explain consistency with at least some of the ice concentration signature in the regional clouds. The measurement issue is that it seems that many of the clouds remain supercooled or perhaps with ice in them at a level that could elude measurement due to detection limits. The initial transition to the point that calculations start in this study is unexplained. This does not greatly harm the paper, since it remains as a mystery for many investigators. However, some simulations performed (CNTRL case primary ice decreased by 10x) make it clear that BR would not occur in clouds with lower primary ice concentrations. This deserves some mention/discussion in the main manuscript. There must be "enough" ice, enough apparently being 0.1 per liter in the temperature regime between 0 and -9 °C, and then BR can ensue and, if done properly, explain ice enhancements observed in many cases. But primary ice nucleation appears not capable of providing that starting ice concentrations, based on available information. I reiterate this point in some of the specific comments below.

**Specific Comments**

1) Abstract: Lines 27-28: In what studies has primary ice formation ever been constrained in the region of this study by aerosol measurements? To me it implies that

someone has reliably connected aerosol measurements and ice nucleating particle measurements to primary ice formation measured in the region.

Lines 30-32: To be explicit, the parameterization bridges the gap between measured ice concentrations outside of SIP regions (perhaps, this is not clearly shown) and ice concentrations ultimately achieved in some areas.

Line 33: "Insensitive to uncertainties in one primary ice production parameterization." Or something to that effect. Without a clear understanding of primary ice formation in these clouds, one cannot claim that the uncertainties or concentrations are known. Given that some observations have not been mentioned, I suggest that qualification be made that end-to-end understanding still eludes the field. What the paper shows, and what is not made clear here, is that this is true as long as some other primary or secondary process is able to produce 0.1 per liter ice to begin with. Unfortunately, none is known to exist. It is wired into the model via use of a parameterization that has not been constrained by INP observations in this region.

2) Introduction: Line 53: Suggest "...generation of ice crystals in a cloud that..." ...ensues following primary ice nucleation, but occurs in large excess to ice nucleating particle (INP) concentrations... Otherwise this seems an awkward definition. It does require INPs or pre-existing ice from some nucleation process, but the key factor is that it far exceeds these initial values in concentration.

Line 77-78: Suggest that "it is" and "it" can be removed from this sentence for readability. Also, please note that neither of the referenced studies in this sentence were for the region of study, and the latter should be qualified as a modeling study.

Line 103-104: This statement does not describe the results herein or in Young et al. (2019) in my opinion. They state, my emphasis added, "Under the **assumption that primary ice is suitably represented by the model**, we must enhance SIP by up to an order of magnitude to simulate observed $N_{ice}$."

They did not observe INPs, correct? To be accurate, "...help to explain..."? Also it is important to note that this study will also not use observation of INPs as a basis for any sort of closure on complete understanding. The implementation in the Morrison scheme does require explaining the suitability of the inherent ice nucleation scheme, and so it was appreciated to have an Appendix relating to that.

Line 149-154: ICNCs were substantially higher than what? The model ice? Expectations? Based on what? Just for reference, the ICNCs quoted here for the noted temperature range are not one order of magnitude higher than expectations based on more recently reported INP measurements in the Southern Hemisphere ocean region (references noted above), and are 3 orders of magnitude higher than Bigg (1973) at the nearest temperature of observation. Also, regarding noting the "frequency" of observations of low aerosol concentrations and high ICNCs, are you failing to acknowledge the other publications on the observational studies that indicate the apparently more frequent observations of supercooled water conditions? What I am getting at is being clear on what part of the discrepancies are endeavored to explain here, how one gets to these concentrations and from what level of ice presence. The model predicted values, or the diminishingly low values one might expect from observations of INPs? Is it possible to state what the lower detection limits for ice concentrations are in observations, and how this might affect the mean values or spread in ice concentrations?

3) Modeling Methods General comment on this section: I found it very confusing to have two of the altered simulations refer to alteration of size dependencies, but in one for the limitation to be that crystals exceeded a certain size (FRAG1siz) and the other to required that size be scaled for a very large size to one much smaller (TAKAHsiz). I have no recommendation for describing size effects that are totally different.

Line 226, end of Section 3: The abstract mentions sensitivity tests on primary ice nucleation. I understand that these are given in the Supplemental, but this should perhaps be mentioned here.

4) Results Line 240: The CNTRL simulation underestimates mean ICNC, but it should be made clear that this is because of the misrepresentation of the highest ICNC, correct?

Line 342-344; lines 352-353: This is not really the potential uncertainty in INPs, right? This is the uncertainty for the parameterization. That parameterization does not have representation in samples from either Antarctica nor from marine boundary layer regions. The reason this becomes important is the final statement "...as long as there are enough crystals to initiate this process." If INP number concentrations are exceedingly low, as they appear to be in some recent studies referenced above, is the BR process ever stimulated unless cloud tops get much colder? It seems not. It seems that some other process is needed to explain the presence of something on the order of 0.1 per liter already at all temperatures warmer than -9 ? Based on Fig. S4, I ultimately noticed that the basis of your statement must be in the simulations where INPs were decreased by a 10x factor. In that case, it appears that the BR process does not come into play. This is an important finding in my opinion. It deserves mention as another key result.

5) Conclusions Lines 356-360: The results indicate that if sufficient ice concentrations are present already, or are parameterized to be at a certain level, then BR can do what is stated here. However, primary ice nucleation is not "limited" in this study. It is far in excess of existing INP measurements.

Appendix A

Lines 392-393: The Meyers et al. (1992) formulation for contact freezing is another parameterization of questionable applicability to this region. Ignoring that or whether it contributes at all or not, I am unclear on what is meant by the rates being further weighted by the effective diffusivity of the contact nucle. You mean an altered assumption is made on their size? Or how size impacts scavenging rate? Please explain.

Lines 397-398: If the parameterization for heterogeneous nucleation has been selected

to compare better with in-cloud ice measurements over the Antarctic Peninsula, does that not "fix" the primary nucleation problem, without necessarily knowing if this is disguising other processes that must be at play to achieve such concentrations? Given what I mention already above, you might need to explain that this selection has been made to solve an issue or is perhaps based on some prior inference that these ICNCs must be the INP concentrations, not because it is the most appropriate thing to do. This is discussed in some of the papers that preceded this one, especially the fact that the parameterization selected was not relevant for application on sea spray particles.

Figures

Figure 2: It is only here that some limitation on ice at 0.005 per liter is mentioned. Apparently, this is the minimum ice concentration accessible at some measurement frequency, due to sample volumes I assume. This is nowhere explained, but it should be reiterated here. This limit also sets the limit on what is referred to as supercooled water versus ice. What if this were set at 0.00005 per liter, which may be closer to what could be available as INPs? Yet in this figure, it somehow looks like there is a threshold around 0.2 per liter. Why is that?

Supporting Information

Text S4: The statement here about the uncertainty in the parameterization used is correct, unlike the inference in the main manuscript. But again here, the 0.005 per liter threshold is mentioned in how it impacts the results of decreasing primary ice nucleation. Does this threshold need to exist? Again, it needs explanation, somewhere in this paper.

Figure S1 and others: Is there a real reason why the scale needs to change? I was quite confused by these figures until I read the last statement of each caption.

**References**

McCluskey, C. S., Hill, T. C. J. , Humphries, R. S., Rauker, A. M., Moreau, S., Strutton, P. G., Chambers, S. D., Williams, A. G., McRobert, I., Ward, J., Keywood, M. D., Harnwell, J., Ponsonby, W., Loh , Z.M., Krummel, P. B., Protat, A., Kreidenweis, S.M., and DeMott, P.J.: Observations of ice nucleating particles over Southern Ocean waters. *Geophysical Research Letters*, 45, 11,989–11,997. https://doi.org/10.1029/2018GL079981.

Schmale, J., Baccarini, A., Thurnherr, I., Henning, S., Efraim, A., Regayre, L., Bolas, C., Hartmann, M.,Welti, A., Lehtipalo, K., Aemisegger, F., Tatzelt, C., Landwehr, S., Modini, R. L., Tummon, F., Johnson, J., Harris, N., Schnaiter, M., Toffoli, A., Derkani, M., Bukowiecki, 35 N., Stratmann, F., Dommen, J., Baltensperger, U., Wernli, H., Rosenfeld, D., Gysel-Beer, M., and Carslaw, K.: Overview of the Antarctic Circumnavigation Expedition: Study of Preindustrial-like Aerosols and Their Climate Effects (ACE-SPACE), *Bull. Amer. Meteor. Soc.*, 100 (11): 2260–2283, DOI:10.1175/BAMS-D-18-0187.1, 2019.

Welti, A., Bigg, E. K., DeMott, P. J., Gong, X., Hartmann, M., Harvey, M., Henning, S., Herenz, P., Hill, T. C. J., Hornblow, B., Leck, C., Löffler, M., McCluskey, C. S., Rauker, A. M., Schmale, J., Tatzelt, C., van Pinxteren, M., and Stratmann, F.: Ship-based measurements of ice nuclei concentrations over the Arctic, Atlantic, Pacific and Southern Ocean, *Atmos. Chem. Phys. Discuss.*, https://doi.org/10.5194/acp-2020-466, in review, 2020.

---

## Author Comment (AC1) · 20 Aug 2020

**ANSWER TO REVIEWER 1**

**General comments:**

The study simulates an Antarctic cloud over the coastal Antarctic and the Weddell Sea in November-December 2015. Secondary ice production from the break-up of collisions between ice particles is added to Weather and Research Forecasting model. The model simulated results are compared to extensive measurements from airborne and ground-based instruments. Their results indicated break-up of collisions between ice particles could account for enhanced ice number in the pristine Antarctic atmosphere, and these results are insensitive to uncertainties in primary ice production. I think the study will be publishable after the following comments are addressed.

We are grateful to the reviewer for his/her comments that have helped us improve our manuscript.

**Specific comments:**

1. My main concern is the experimental design. The control simulation is stated as using the default Morrison scheme (Line 204: "Additionally to the control (CNTRL) simulation, which corresponds to the default set-up of M05"). The Morrison scheme is described in the literature that includes different types of secondary ice production in it, i.e. the rime splintering (H-M) during snow accretion cloud droplets and snow accretion raindrops. Therefore, either I have misinterpreted and some additional description of the control model configuration is needed, or I think we need an additional set of control runs that have no secondary ice production processes included Until there is a clean experiment with no secondary ice production processes it is difficult to interpret the statements about the impact of secondary ice production.

Thank you for this suggestion. The simulation with no SIP at all was initially included in the Supplementary Information, in the section that the sensitivity to the H-M process was tested. This test was referred as 'CNTRL\_NOHM', which corresponded to a simulation with the default M05 but with H-M deactivated. In the revised text we now refer to this simulation as 'NOSIP'. Moreover, we have included the 'NOSIP' results in Figure 2, to help the readers quantify the impact of secondary ice production by comparing the rest of the simulations to 'NOSIP'.

2. My first concern is about "separation size between ice and snow" The authors mentioned in the manuscript Line 215: "Note that since the separation size between ice and snow in the M05 scheme is 125  $\mu$ m, collisions that include cloud ice do not result in any multiplication in FRAG1siz." The reader may be confused, does Morrison scheme really has a size separation between cloud ice and snow? Does this mean the model does not have cloud ice larger than 125  $\mu$ m, and no snow particles with a diameter smaller than 125  $\mu$ m? Morrison scheme is documented in the literature that includes a threshold size (125  $\mu$ m) for the cloud ice autoconversion process. But this does not mean a size separation between could ice and snow. Based on this, when comparing simulated snow and graupel with observation, does modeled snow and graupel only considering particles

**larger than 80 µm? Or only modeled cloud ice has this threshold in size.**

Thank you for pointing out this mistake. Cloud ice is converted to snow when diameter exceeds  $d_{cs}=125 \ \mu m$ . However since cloud ice (and snow) size spectra are represented with a complete gamma distribution, this means that sizes larger than 125  $\mu m$  are not excluded in the spectra (and, in fact, by the definition of a complete gamma function the sizes mathematically extend to infinity). However, if the characteristic diameter of the cloud ice category is larger than 250  $\mu m$  then all cloud ice is converted to snow in M05. Since break-up in FRAG1siz is allowed only if the characteristic diameter of the particle that undergoes break up is larger than 300  $\mu m$ , collisions with cloud ice do not contribute to multiplication. This statement is now corrected in the revised text (lines 219-221).

Regarding the modeled output plotted in Figures 2-3: when outputs are compared to observations, then only particles larger than 80  $\mu$ m are accounted for consistency with the cloud phase detection limit in the observations; this holds for model data plotted in Fig. 2 and Fig. 3a,c,d. In Figure 3b, graupel is not compared to observations, so we plot the whole graupel spectrum to show how limited its concentration is. This is better explained in the revised text (lines 282-290) and Figure caption to avoid confusion.

- 3. Related to the second concern, the third concern is related to the comparison between observation and model results. The authors mentioned that "consistency with M05, the threshold size separating measured cloud ice from snow is set to 125 µm" Usually, when comparing model ice with observation, we added different types of modeled solid particles together, then using the total mass and total number to compare with the observed IWC and ice number concentration. Because it is hard to tell cloud ice from snow in observation data. I suggested using a similar method when compare the modeled ice number with observation. We have corrected the adapted threshold in this figure; this is now set to 250 µm for consistency with the microphysics scheme assumptions (see answer above). Although it is hard to separate cloud ice/snow in observations, 2DS measurements offer a good indication of the size spectra. Plotting cloud ice and snow categories separately in this figure is important, because it shows that including break-up in the scheme results in more realistic representation of the microphysical properties. Particularly, it is important to show that it is the concentration of the small particles that is substantially underestimated in the control simulation (more than 2.5 orders of magnitude, see Figure 3a); concentration of large particles (>250 µm) is underpredicted by less than a factor of 1.5 (see Figure 3c). Thus, break-up shifts the modeled size spectra towards smaller values in agreement with observations, which indicates that we are likely capturing the correct mechanisms responsible for the ice microphysical characteristics. This is now discussed in lines 302-307. We have also included a fourth panel in this figure, to show the vertical profiles of total ICNCs.
- 4. The fourth concern is related to parameterization from Phillips et al. (2017) In the sensitivity test, Line 255, "These simulations are referred to as PHIL0.2, PHIL0.3 and PHIL0.4 in the text, where the number indicates the assumed values of Ψ". In the bulk microphysical scheme, snow is referred to as dry snow, with smaller density (prescribed and fixed in the scheme), graupel is rimed ice with larger

density. Setting the same rimed fraction for different collisions is not consistent with the assumptions in the microphysics scheme. I suggested the uses different rimed fraction for snow-snow and graupel-graupel collision. The rimed fraction could be changed in the sensitivity test, but in one simulation, the rimed fraction for the collision between snow-snow should not larger than rimed fraction for the collision between graupel-graupel.

We apologize for this confusion. In the Phillips parameterization, graupel particles are highly breakable as  $\Psi$  is assumed to be larger than 0.5; the fragment generation does not depend on  $\Psi$  and is only a function of temperature (see equations at line 622 in Appendix B). In contrast,  $\Psi$  is important for cloud ice/snow and that's why fragment generation directly depends on  $\Psi$  for these ice categories (equations in line 609). The parameterization assumes that  $\Psi$  for these ice types is less than 0.5, lower than for graupel, in agreement with the reviewer's comment.

The different assumptions regarding the rimed fraction of graupel and the rest of the ice categories are now discussed in detail to avoid confusion (lines 229-235 in the revised text). It is also explicitly stated that variations in  $\Psi$  only affect the break-up efficiency of cloud ice/ snow.

5. The last one is related to radiation. Does cloud microphysical properties couple with radiation transform code? It is interesting to see the model has a larger ice number and IWC, after the implementation of the secondary ice process, but the longwave radiation does not change accordingly. How about the effective radius of the ice particle? Does it change after the model has secondary ice production in it?

This is an excellent point. Only the mass mixing ratios of liquid and ice particles are transferred to the radiation scheme and drive the differences reported in Table 1. No information on effective radius is directly passed to RRTMG scheme. This is now clarified in the text (lines 323-325).

**Technical corrections**

1. Line 59 "Lachlan-Cope et al., 2016; Wexx et al.,". Wexx ->Wex?

Although the statement was more general about polar clouds, we have removed the reference 'Wex et al' since this concerns Arctic clouds

- 2. Line 96 be -> been corrected
- 3. Code and data availability: the link authors provide does not link to the measurement data, please upload the data.

We apologize that the direct link did not work in the previous manuscript. However, the provided URL address is valid. Hopefully, the link works fine in the revised version (in any case copy-pasting the address in a browser should definitely work)

**Comments embedded in the pdf version of the manuscript**

The reviewer points out that strong riming is the foundation of the hallet-mossop process. This sounds somewhat contradicting to our findings regarding the fact that only simulations that assume a high rimed fraction for the particles that undergo break-up during ice particle collisions produce realistic ICNCs, while at the same time H-M remains inactive.

While in Phillips parameterization for break-up, rimed fraction is explicitly considered, this is not the case for H-M. To make sure that only sufficiently rimed particles contribute to multiplication, H-M is activated only if the ice particle mixing ratio exceeds 0.1 g kg-1. However, this threshold is ad-hoc and is tuned for midlatitude clouds, but is hardly exceeded in polar clouds. In the former submission we discarded the liquid thresholds in the parameterization and found no impact on the results. In the revised manuscript, we removed both liquid and ice thresholds and allowed H-M to be active over the whole droplet and snow/graupel spectrum. This enhances the H-M efficiency by a factor of 3 but still cannot reproduce observed mean ICNCs. This set-up overestimates H-M efficiency, as size limitations have been reported in laboratory studies, but the existing thresholds in M05 should be refined for polar clouds. The whole text has been moved from the Supplementary Material to the main manuscript (section 4.3).

---

## Author Comment (AC2) · 20 Aug 2020

**ANSWER TO REVIEWER 3**

**General Comments**

This is a good paper overall, in its seeking to explore the role of different secondary ice formation mechanisms active under conditions present in clouds over the focused region near Antarctica. To the extent that I can tell, the findings with regard to BR parameterizations are solid and enlightening regarding the necessary conditions (especially heavy riming) for it to occur and the need for formulations of BR that account for the requisite size of colliding particles

We are grateful for the many insightful comments that have helped us improve our manuscript.

This paper was less forward and unsatisfying in conveying that end-to-end prediction of secondary ice formation actually still eludes the community. This is my inference from this paper and from the literature of the last few years represented by some of the team on this paper. This relates to the assumptions on primary nucleation, and the concentrations of ice that exists outside of regions where strong SIP ensues. There is still a gap here, one that is not really discussed, although this study seems to close it some. By this I mean that the authors have proposed to explain how ice concentrations go from values of 0.1 per liter to ones enhanced up to 100 times, and under conditions that the Hallett-Mossop process cannot explain such enhancement. The results cannot explain the inconsistency between recent INP measurements over the Southern Ocean or near Antarctica (e.g., McCluskey et al., 2018; Schmale et al., 2019; Welti et al., 2020) and the fact that Litkowski et al. (2017) required use of the DeMott et al. (2010) parameterization (with qualifications about its likely inapplicability to the region) in order to explain consistency with at least some of the ice concentration signature in the regional clouds.

The summer coastal low-level Antarctic clouds are dominated by supercooled liquid droplets while ice occurs in isolated large ice patches. Our statistics that focus on cloud ice properties by default refer to the ice-containing regions of the cloud. This indeed was not clear in the initial version of the manuscript and now is explicitly mentioned in the abstract, case description and conclusion section.

Although no primary ice scheme is likely suitable for polar conditions, DeMott et al. (2010) still performs better than all other parameterizations available in WRF (and generally in models with no prognostic aerosols). This was discussed in Listowski et al. (2017). Sensitivity tests however are considered to address the factor of 10 uncertainty in the chosen parameterization, but we acknowledge that the results of these tests were not emphasized enough in the main manuscript and particularly the large sensitivity to INP reductions. For this reason the whole section on primary ice nucleation has been moved from the Supplementary Material to the main text (section 4.4). The fact that a minimum primary ICNC concentration of  $0.1 \text{ L}^{-1}$  is needed to initiate break-up is now explicitly stated in both 'abstract' and 'conclusions' section. Potential suggestions on how this requirement might be met are offered in sections 4.4 and 5 (e.g. ice seeding as likely in our case, transport of terrestrial aerosols, more efficient H-M).

Moreover, the deficiencies of the applied parameterizations, such as the fact that they

have been neither developed or calibrated for polar conditions, are also explicitly mentioned in the revised text (lines 399-404, 499-500).

The measurement issue is that it seems that many of the clouds remain supercooled or perhaps with ice in them at a level that could elude measurement due to detection limits. The initial transition to the point that calculations start in this study is unexplained. This does not greatly harm the paper, since it remains as a mystery for many investigators. However, some simulations performed (CNTRL case primary ice decreased by 10x) make it clear that BR would not occur in clouds with lower primary ice concentrations. This deserves some mention/discussion in the main manuscript. There must be "enough" ice, enough apparently being 0.1 per liter in the temperature regime between 0 and -9°C, and then BR can ensue and, if done properly, explain ice enhancements observed in many cases. But primary ice nucleation appears not capable of providing that starting ice concentrations, based on available information. I reiterate this point in some of the specific comments below.

The sensitivity in low INP conditions is now discussed more extensively in section 4.4 and the fact that there might be a 'triggering' primary ICNC threshold that can explain the observed ice patches within the predominately liquid clouds. Furthermore we discuss possible processes that can provide the necessary conditions to meet this requirement (e.g. ice seeding in the particular case). This finding is also repeated in the 'conclusions' section.

**Specific Comments**

**Abstract:**

**Lines 27-28:** In what studies has primary ice formation ever been constrained in the region of this study by aerosol measurements? To me it implies that someone has reliably connected aerosol measurements and ice nucleating particle measurements to primary ice formation measured in the region.

The abstract has been modified substantially and this sentence has been removed.

**Lines 30-32:** To be explicit, the parameterization bridges the gap between measured ice concentrations outside of SIP regions (perhaps, this is not clearly shown) and ice concentrations ultimately achieved in some areas.

By default ice properties are only calculated with the observed ice patches where SIP seems to occur. To avoid any confusion this is now explicitly stated in the abstract along with the fact that these ice patches are observed within predominantly liquid cloud layers.

Line 33: "Insensitive to uncertainties in one primary ice production parameterization." Or something to that effect. Without a clear understanding of primary ice formation in these clouds, one cannot claim that the uncertainties or concentrations are known. Given that some observations have not been mentioned, I suggest that qualification be made that end-to-end understanding still eludes the field. What the paper shows, and what is not made clear here, is that this is true as long as some other primary or secondary process is able to produce 0.1 per liter ice to begin with. Unfortunately, none is known to exist. It is wired into the model via use of a parameterization that has not been constrained by INP observations in this region.

We now make clear in both abstract and 'conclusions' section that a minimum

primary ice concentration of  $0.1 \text{ L}^{-1}$  is needed to initiate break-up. Potential contributions to this minimum value are also mentioned in section 4.4. and 5.

2) Introduction: Line 53: Suggest ": : :generation of ice crystals in a cloud that: : :" ...ensues following primary ice nucleation, but occurs in large excess to ice nucleating particle (INP) concentrations: : Otherwise this seems an awkward definition. It does require INPs or pre-existing ice from some nucleation process, but the key factor is that it far exceeds these initial values in concentration. The whole paragraph has been revised (lines 51-55).

Line 77-78: Suggest that "it is" and "it" can be removed from this sentence for readability. Also, please note that neither of the referenced studies in this sentence were for the region of study, and the latter should be qualified as a modeling study. Thank you, corrected

Line 103-104: This statement does not describe the results herein or in Young et al.(2019) in my opinion. They state, my emphasis added, "Under the assumption that primary ice is suitably represented by the model, we must enhance SIP by up to an order of magnitude to simulate observed Nice:" They did not observe INPs, correct? To be accurate, ": : :help to explain: : :"? Also it is important to note that this study will also not use observation of INPs as a basis for any sort of closure on complete understanding. The implementation in the Morrison scheme does require explaining the suitability of the inherent ice nucleation scheme, and so it was appreciated to have an Appendix relating to that.

We have modified this statement, which was inaccurate, by referring now to the discrepancy between observed and modeled ICNCs in line 101 (instead of INPs and ICNCs).

Line 149-154: ICNCs were substantially higher than what? The model ice? Expectations? Based on what? Just for reference, the ICNCs quoted here for the noted temperature range are not one order of magnitude higher than expectations based on more recently reported INP measurements in the Southern Hemisphere ocean region (references noted above), and are 3 orders of magnitude higher than Bigg (1973) at the nearest temperature of observation. Also, regarding noting the "frequency" of observations of low aerosol concentrations and high ICNCs, are you failing to acknowledge the other publications on the observational studies that indicate the apparently more frequent observations of supercooled water conditions? What I am getting at is being clear on what part of the discrepancies are endeavored to explain here, how one gets to these concentrations and from what level of ice presence. The model predicted values, or the diminishingly low values one might expect from observations of INPs? Is it possible to state what the lower detection limits for ice concentrations are in observations, and how this might affect the mean values or spread in ice concentrations?

Although clearly INPs can be substantially lower over the Southern Ocean, we do not have INP measurements for the particular case. For this reason we only state the aerosol concentrations available to indicate how clean is the measured atmosphere. Assuming that the average of observed INP from the literature is more representative for our case than INP diagnosed from direct aerosol observations (with uncertainty of one order of magnitude) is also subject to considerable uncertainty. Considering the high ICNC conditions in the observed cloud ice patches, surface INP measurements may not account for ice-nucleating particles transported above/within the cloud.

When calculating ICNC statistics, by default this estimate concerns the ice-containing regions of the cloud. However, we acknowledge that this was not explicitly discussed in the previous version of the paper.

Regarding detection limits, the 2DS is a single particle instrument: it measures all particles that pass through its sample volume with size larger than ~10  $\mu$ m. However phase identification cannot be conducted for particles with sizes lower than 80  $\mu$ m. The 2DS sample volume depends on particle size and the data integration period. For example at 300 um the sample volume is 3.7 L/s. For 1 count measured within 1-sec averaging window, this equals a concentration of 0.27 L-1 ( 1 count/ (3.7 L/s \* 1s )). If a 10-sec window was chosen, then this value would go down to 0.027 L-1. There is an uncertainty in the concentration due to the counting statistics (1/sqrt(counts)). For 1 count the uncertainty due to counting statistics is 100 %. This is now explained in section 2.1, lines 121-125. The total uncertainty in ICNCs is even larger but cannot be determined. However, what the applied thresholds ensure is that calculations concern ice patches, not liquid-only regions (line 243-244), and that this is consistent for both observations and model.

**3) Modeling Methods General comment on this section**: I found it very confusing to have two of the altered simulations refer to alteration of size dependencies, but in one for the limitation to be that crystals exceeded a certain size (FRAG1siz) and the other to required that size be scaled for a very large size to one much smaller (TAKAHsiz). I have no recommendation for describing size effects that are totally different.

We changed TAKAHsiz to TAKAHsc (to indicate scaling for size)

Line 226, end of Section 3: The abstract mentions sensitivity tests on primary ice nucleation. I understand that these are given in the Supplemental, but this should perhaps be mentioned here.

The whole primary ice section has been moved from the Supplementary Information to the main manuscript as section 4.4.

**4) Results**

**Line 240:** The CNTRL simulation underestimates mean ICNC, but it should be made clear that this is because of the misrepresentation of the highest ICNC, correct?

Yes, as now stated in lines 268-269 CNTRL cannot reproduce the whole spectrum and largely underestimates the frequency of ICNCs  $> 1 L^{-1}$

Line 342-344; lines 352-353: This is not really the potential uncertainty in INPs, right? This is the uncertainty for the parameterization. That parameterization does not have representation in samples from either Antarctica nor from marine boundary layer regions. The reason this becomes important is the final statement ": : :as long as there are enough crystals to initiate this process."

The discussion in section 4.4 now addresses the fact that the parameterization is not constrained based on Antarctic measurements. The fact that a minimum primary concentration of  $0.1 \text{ L}^{-1}$  is required to initiate BR is now emphasized throughout the whole manuscript.

If INP number concentrations are exceedingly low, as they appear to be in some recent studies referenced above, is the BR process ever stimulated unless cloud tops get much colder? It seems not. It seems that some other process is needed to explain the presence of something on the order of 0.1 per liter already at all temperatures warmer than -9? Based on Fig. S4, I ultimately noticed that the basis of your statement must be in the simulations where INPs were decreased by a 10x factor. In that case, it appears that the BR process does not come into play. This is an important finding in my opinion. It deserves mention as another key result.

This is now mentioned as a key result. We also suggest processes that might result in conditions favorable for BR (e.g. ice seeding, aerosol transport). It is likely that understanding the interactions between such processes might explain the reason why large ice patches with substantially enhanced ICNCs are observed when supercooled liquid cloud conditions generally dominate.

**5)** Conclusions Lines **356-360**: The results indicate that if sufficient ice concentrations are present already, or are parameterized to be at a certain level, then BR can do what is stated here. However, primary ice nucleation is not "limited" in this study. It is far in excess of existing INP measurements

The conclusion section now emphasizes that certain primary ice conditions are required to initiate BR. We also explicitly state that such INP conditions are not frequently found over Southern Ocean, which is likely the reason why supercooled droplets dominate in these clouds.

**Appendix A**

**Lines 392-393:** The Meyers et al. (1992) formulation for contact freezing is another parameterization of questionable applicability to this region. Ignoring that or whether it contributes at all or not, I am unclear on what is meant by the rates being further weighted by the effective diffusivity of the contact nuclei. You mean an altered assumption is made on their size? Or how size impacts scavenging rate? Please explain.

The effective diffusivity of the contact nuclei is estimated based on Brownian motion as (similar to Young 1974):  $D_{ap} = R T (6 p r_i N_A m)^{-1} [1 + 0.0737 T (2880P)^{-1} r_i^{-1}]$ , where *R* is the universal gas constant,  $N_A$  is Avogadro's number, *m* is the dynamic viscosity of air, *T* is the temperature, *P* is the air pressure, and the radius of ice nuclei  $r_i$  is assumed to be 1 x 10-7 m. The factor in the brackets [] is a correction factor accounting for the mean free path of air molecules relative to the size of the ice nuclei (all units are MKS). This information is now provided in the revised manuscript in lines 488-493

Lines 397-398: If the parameterization for heterogeneous nucleation has been selected to compare better with in-cloud ice measurements over the Antarctic Peninsula, does that not "fix" the primary nucleation problem, without necessarily knowing if this is disguising other processes that must be at play to achieve such concentrations? Given what I mention already above, you might need to explain that this selection has been made to solve an issue or is perhaps based on some prior inference that these ICNCs must be the INP concentrations, not because it is the most appropriate thing to do. This is discussed in some of the papers that preceded this one, especially the fact that the parameterization selected was not relevant for application

**on sea spray particles.**

Previous comparisons of this scheme and other primary ice nucleation schemes available in WRF for Antarctic Clouds have shown that the Cooper parameterization performs worse (Listowski et al. 2017). This is also the case for Arctic clouds (Young et al. 2017). Nevertheless, DeMott's and Cooper's schemes produce similar primary ice over the temperature range covered by the observations, but the latter provides unrealistically high values at lower temperatures (see Young et al. 2017 and Supplementary Information of Young et al. 2019). This is now explained in Appendix A (lines 496-504).

**Figures**

**Figure 2:** It is only here that some limitation on ice at 0.005 per liter is mentioned. Apparently, this is the minimum ice concentration accessible at some measurement frequency, due to sample volumes I assume. This is nowhere explained, but it should be reiterated here. This limit also sets the limit on what is referred to as supercooled water versus ice. What if this were set at 0.00005 per liter, which may be closer to what could be available as INPs? Yet in this figure, it somehow looks like there is a threshold around 0.2 per liter. Why is that?

1 count measured by 2DS within 1-sec averaging window corresponds to a concentration of 0.27 L-1. This was the minimum concentration in our previous manuscript. Discrepancies between the data presented in the previous manuscript version and Young et al. (2019) are due to different data processing methods. In our case, we first applied the 0.005 L-1 threshold to our dataset and then interpolated ICNCs to the time resolution of temperature measurements. Young et al. (2019) interpolated ICNCs first and then applied the cut-off ICNC threshold. Their interpolation resulted sometimes in lower values, as interpolation between non-ice containing and ice containing regions occurred. The minimum value in their data however is  $0.007 L^{-1}$ , still higher than the applied threshold. In other words applying even lower thresholds would not impact observational statistics. Nevertheless, for consistency between the two papers we now use the exact same data-processing method as in Young et al. (2019) and describe this in lines 242-244.

As far as modeled ICNCs are concerned, ice processes are calculated only if meaningful ice content is present ( $Q_i > 10^{-8} \text{ kg kg}^{-1}$ ). The minimum ICNC (sizes >80 µm) corresponding to these Qi conditions found in the CNTRL model dataset is 0.0008 L-1. This means that no meaningful ice is produced for INPs ~ 0.00005 L-1, which corresponds to supercooled liquid-only conditions in the model. Considering the high frequency of liquid-only clouds in the region, this is the expected behaviour.

**Supporting Information**

Text S4: The statement here about the uncertainty in the parameterization used is correct, unlike the inference in the main manuscript. But again here, the 0.005 per liter threshold is mentioned in how it impacts the results of decreasing primary ice nucleation. Does this threshold need to exist? Again, it needs explanation, somewhere in this paper.

Yes, this threshold separates ice patches from liquid-only regions of the cloud. This is now stated in lines 243-244. Including non-ice containing clouds in the calculation of ice microphysical properties would cause a significant bias in the results.

**Figure S1 and others:** Is there a real reason why the scale needs to change? I was quite confused by these figures until I read the last statement of each caption.

Supplementary Figures have been moved to the main text (Figure 5 and 6). The same scaling is now used in all plots to avoid confusion.

**Reference:**

Young, K. C., 1974: The Role of Contact Nucleation in Ice Phase Initiation in Clouds. J. Atmos. Sci., 31, 768–776, https://doi.org/10.1175/1520-0469(1974)031<0768:TROCNI>2.0.CO;2.

Young, G., Connolly, P. J., Jones, H. M., and Choularton, T. W.: Microphysical sensitivity of coupled springtime Arctic stratocumulus to modelled primary ice over the ice pack, marginal ice, and ocean, Atmos. Chem. Phys., 17, 4209–4227, https://doi.org/10.5194/acp-17-4209-2017, 2017.

---

## Referee Report (RR1)

Followup review of "Secondary ice production in summer clouds over the Antarctic coast: an underappreciated process in atmospheric models" by Sotiropoulou et al.

**General Comments**

The authors have done a very nice job of pulling material into the main manuscript or otherwise clarifying statements, measurements and modeling parameters in responding to my original comments. I have just a few suggestions in hopes to make crystal clear what advances are made in this paper versus what pieces of the puzzle remain for investigation. In particular, the paper still somewhat dances around the point that primary ice nucleation based on current information appears not capable of providing the requisite starting ice concentrations for the BR process to then provide the critical secondary mechanism needed to explain ultimate ice concentrations achieved in pockets of clouds over the region. A qualitative word, "meaningful" is used to describe ice concentrations of relevance, and yet, what has been measured so far in the region of interest is then not "meaningful." Just a few word changes will clarify this point.

**Specific Comments**

1) Abstract, line 36: I suggest "necessary and sufficient". This seems to me to be true. Sufficient alone implies to one that this is not difficult to explain. I understand that some possible explanations have been provided, but there remain significant efforts to prove any such points. I do not mean to detract from the significant results of the paper, only to emphasize mildly that this is a missing piece.

2) Section 4.4, lines 399-404: I believe that it is not an issue of calibration or even potentially missing INP sources, as this presently suggests. I can suggest the emphasized words in "None of the utilized primary ice nucleation parameterizations were **developed or are likely representative for**…". The following sentence appears irrelevant and should be removed. The pristine or limited nature of the INP populations in the region is the important point, and no new source is likely to explain any missing populations. If trying to say that new parameterizations will be needed to accurately represent sources, that would be fine, but it is not articulated that way in this revised wording.

3) Paragraph starting on line 440: This is excellent. One minor note on line 437. Terrestrial INPs are "assumed to be" or "expected to be" higher? The noted model study reference discusses contrasts between land and oceanic INP concentrations, but does not demonstrate that Antarctica is a specific elevated source of INPs.

4) Conclusions, lines 455-457: Again the same point as above. Suggest (new words emphasized in bold and others crossed out) "…such conditions could  be achieved through ice seeding (as likely happens in the examined case) or through INP transport from the Antarctic continent, where INP concentrations are  **predicted to be** higher (Vergara-Temprado et al., 2018). **These points remain for future confirmation.**"

5) Appendix A, line 500: Suggest to replace "have been calibrated" with "were not developed to be representative of". Also, add "parameterizations" after Cooper and DeMott.

---

## Author Response (AR2)

**Answer to the reviewers :**

We are grateful to the reviewers for very detailed comments that help clarify several aspects of the manuscript. Reviewers' comments are highlighted in red, while our response follows in black.

**Reviewer 2**

Minor suggestion for revision: \Psi is defined as rimed fraction in line 92; it would be helpful to repeat this definition when the values of \Psi are discussed in lines 228 ff.

 $\Psi$  is now defined as rimed fraction also in line 231 in the revised text.

**Reviewer 3.**

**General Comments**

The authors have done a very nice job of pulling material into the main manuscript or otherwise clarifying statements, measurements and modeling parameters in responding to my original comments. I have just a few suggestions in hopes to make crystal clear what advances are made in this paper versus what pieces of the puzzle remain for investigation. In particular, the paper still somewhat dances around the point that primary ice nucleation based on current information appears not capable of providing the requisite starting ice concentrations for the BR process to then provide the critical secondary mechanism needed to explain ultimate ice concentrations achieved in pockets of clouds over the region. A qualitative word, "meaningful" is used to describe ice concentrations of relevance, and yet, what has been measured so far in the region of interest is then not "meaningful." Just a few word changes will clarify this point.

**Specific Comments**

1) Abstract, line 36: I suggest "necessary and sufficient". This seems to me to be true. Sufficient alone implies to one that this is not difficult to explain. I understand that some possible explanations have been provided, but there remain significant efforts to prove any such points. I do not mean to detract from the significant results of the paper, only to emphasize mildly that this is a missing piece.

"necessary and sufficient" has now been added to the abstract

2) Section 4.4, lines 399-404: I believe that it is not an issue of calibration or even potentially missing INP sources, as this presently suggests. I can suggest the emphasized words in "None of the utilized primary ice nucleation parameterizations were **developed or are likely representative for**...". The following sentence appears

irrelevant and should be removed. The pristine or limited nature of the INP populations in the region is the important point, and no new source is likely to explain any missing populations. If trying to say that new parameterizations will be needed to accurately represent sources, that would be fine, but it is not articulated that way in this revised wording.

It is now clarified that none of the utilized primary ice nucleation parameterizations 'are likely representative for' the pristine conditions encountered over the highlatitude Southern Ocean. The following sentence has been removed, as suggested by the reviewer, to avoid confusion

3) Paragraph starting on line 440: This is excellent. One minor note on line 437. Terrestrial INPs are "assumed to be" or "expected to be" higher? The noted model study reference discusses contrasts between land and oceanic INP concentrations, but does not demonstrate that Antarctica is a specific elevated source of INPs.

It is now clarified that terrestrial INPs are "expected to be" higher

4) Conclusions, lines 455-457: Again the same point as above. Suggest (new words emphasized in bold and others crossed out) "...such conditions could likely be achieved through ice seeding (as likely happens in the examined case) or through INP transport from the Antarctic continent, where INP concentrations are generally **predicted to be** higher (Vergara-Temprado et al., 2018). **These points remain for future confirmation.**"

Rephrased exactly as suggested by the reviewer

5) Appendix A, line 500: Suggest to replace "have been calibrated" with "were not developed to be representative of". Also, add "parameterizations" after Cooper and DeMott.

[revised manuscript text omitted]

**Figures:**

975 Figure 1: Map of Antarctic domains. Colors indicate terrain heights (green to yellow) and sea-ice concentrations (blue to white), whereas the purple contours correspond to 500 hPa geopotential heights from the CNTRL simulation at 18:00 UTC, 27 November 2015. The black solid line delimits the 1-km horizontal grid spacing domain, while the dashed one outlines the subset of the nest used for direct comparison with the aircraft data. Orange and blue lines indicate the flight tracks, while the red circle represents Halley station. The small figure in the top right corner indicates the location of the 1-km horizontal grid spacing domain relative to the Antarctic continent.

31